# GAUSSIAN PROCESS PRIORS FOR BOUNDARY VALUE PROBLEMS OF LINEAR PARTIAL DIFFERENTIAL EQUATIONS

## ABSTRACT

Working with systems of partial differential equations (PDEs) is a fundamental task in computational science. Well-posed systems are addressed by numerical solvers or neural operators, whereas systems described by data are often addressed by PINNs or Gaussian processes. In this work, we propose Boundary Ehrenpreis–Palamodov Gaussian Processes (B-EPGPs), a novel probabilistic framework for constructing GP priors that satisfy both general systems of linear PDEs with constant coefficients and linear boundary conditions and can be conditioned on a finite data set. The Ehrenpreis–Palamodov theorem provides the functional form of the solution, but leaves parameters of this functional form unknown. An optimization finds these parameters and we provide an algorithm to modify this function form to satisfy boundary conditions. We explicitly construct GP priors for representative PDE systems with practical boundary conditions. Formal proofs of correctness are provided and empirical results demonstrating significant accuracy and computational resource improvements over state-of-the-art approaches.

## 1 INTRODUCTION

Classically, systems of partial differential equations (PDEs) have been *solved* using numerical methods, which require the solution to be uniquely determined. This uniqueness is typically ensured by prescribing a sufficient number of initial or boundary conditions. Neural operators adopt a similar strategy, and learn the evolution of PDE solutions, mapping the system state at time $t$ to the state at time $t + 1$, given appropriate initial conditions. While such methods significantly reduce computational cost, their accuracy falls short of that achieved by traditional numerical solvers.

A second line of research in machine learning abandons the requirement of uniqueness and instead seeks a "good" approximate solution that fits observed data, i.e. *regression* inside of the solution set of PDEs. This paper falls in this line of work. A prominent example is physics-informed neural networks (PINNs) (Raissi et al., 2019), which combine standard regression loss with an additional penalty for violating the PDE at sampled points in the domain. Many Gaussian process (GP) models for linear PDEs also follow this approach.

GPs (Rasmussen & Williams, 2006) are a standard choice for functional priors, offering robust regression with few data points and calibrated uncertainty estimates. Their covariance structure can encode linear properties, such as derivatives (Swain et al., 2016; Harrington et al., 2016), enabling the construction of GPs whose realizations lie in the solution set of linear PDEs with constant coefficients (Lange-Hegermann, 2018; Jidling et al., 2017; Macêdo & Castro, 2008; Scheuerer & Schlather, 2012; Wahlström et al., 2013; Solin et al., 2018; Jidling et al., 2018; Särkkä, 2011), provided the system is controllable, i.e., admits potentials. This construction, relying on Gröbner bases for multivariate polynomial rings, was extended beyond controllable systems (Harkonen et al., 2023). These methods yield machine learning models whose predictions are not merely physics-*informed*, but physics *constrained*: all regression functions are exact solutions of the PDE system. As a result, they surpass PINNs in accuracy by several orders of magnitude.

In practice, PDE systems are typically accompanied by both data and (initial or) boundary conditions. For controllable systems, such boundary conditions can be incorporated into GP models using Gröbner bases over the Weyl algebra (Lange-Hegermann, 2021). However, this approach does not extend to

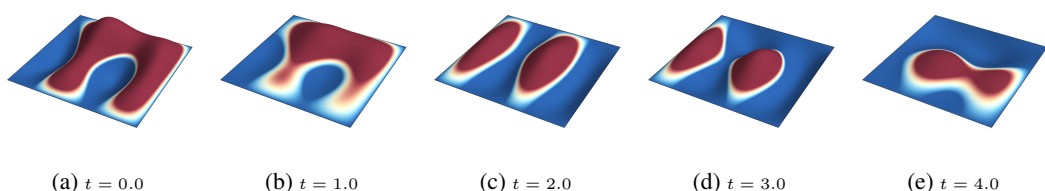

(a) $t = 0.0$     (b) $t = 1.0$     (c) $t = 2.0$     (d) $t = 3.0$     (e) $t = 4.0$

Figure 1: Consider the 2D wave equation $\partial_t^2 u - \partial_x^2 u - \partial_y^2 u = 0$ with zero boundary conditions on a rectangular spatial domain. This figure shows a sample drawn from our B-EPGP construction. Every sample analytically satisfies both the differential equation and the boundary conditions, and can be conditioned on any finite set of observations.

general linear PDEs with constant coefficients. In this paper, we introduce an algorithmic construction of GP priors—called Boundary Ehrenpreis–Palamodov Gaussian Processes (B-EPGPs)—that reside within the solution space of *any* such system, including linear boundary conditions. B-EPGP builds on the infinite-dimensional "basis" provided by the Ehrenpreis–Palamodov theorem and transforms it into a "basis" of solutions that also satisfy the boundary conditions. Our main contributions are:

1. We present a general algorithmic construction of a novel model class: GP priors for solutions of linear PDE systems with constant coefficients and linear boundary conditions.

2. We provide explicit constructions of such priors for several representative PDEs with practical boundary setups.

3. We thoroughly prove correctness and convergence of our approach, in particular that these GP priors have realizations that are dense in the PDE solution.

While our approach is mainly a regression model, we empirically demonstrate that it significantly outperforms state-of-the-art neural operator models by orders of magnitude in accuracy and computation time. Our approach scales mildly with the dimension, handling e.g. the (3+1)D wave equation with minimal boundary data, avoiding the exponential boundary discretization cost of traditional solvers.

## 2 GAUSSIAN PROCESS PRIORS FROM THE EHRENPREIS–PALAMODOV THEOREM

A key distinction between linear and nonlinear ODEs is that solutions to linear systems with constant coefficients are linear combinations of exponential-polynomial functions. For example, the ODE $y''' - 3y' + 2y = 0$ has linear combinations of $e^x$, $xe^x$, and $e^{-2x}$ as solutions, where the "frequencies" 1 and $-2$ are the roots of the *characteristic polynomial* $z^3 - 3z + 2 = (z-1)^2(z+2)$.

Remarkably, this structure extends to systems of linear PDEs with constant coefficients. Consider the 1D heat equation $\partial_1 u - \partial_2^2 u = 0$. Its exponential-polynomial solutions take the form $e^{x_1 z_1 + x_2 z_2}$, where the frequency vector $(z_1, z_2)$ satisfies the characteristic equation $z_1 - z_2^2 = 0$. This equation arises substituting each derivative $\partial_i$ with a complex variable $z_i$, akin to a Fourier transform. The set of such $z$ defines the *characteristic variety*, here given by $V = \{(z_2^2, z_2) : z_2 \in \mathbb{C}\}$. We state the special case of the Ehrenpreis–Palamodov theorem for irreducible characteristic varieties without multiplicities. We refer to Appendix F for the general case, which involves linear combinations over the components of the variety and correction terms (as the factor $x$ in $xe^x$ above) for multiplicities. We say that a set $\mathcal{S}$ is a *Finitely-Approximating Set (FAS)* of a complex Fréchet space $\mathcal{X}$ if the complex linear span of $\mathcal{S}$ is dense in $\mathcal{X}$, e.g. $\mathcal{X} = \ker_{C^\infty(\Omega)} A = \{u \in C^\infty(\Omega, \mathbb{C}) : A(\partial)u = 0\}$, where $A$ is a linear PDE operator and $\partial = (\partial_1, \ldots, \partial_n)$ denote *real* partial derivatives. See also Appendix F.

**Theorem 2.1.** *Let $\Omega \subset \mathbb{R}^n$ be an open convex set. Let $A \in \mathbb{C}[\partial]$ and $V = \{z \in \mathbb{C}^n : A(z) = 0\}$. Suppose $V$ is irreducible and has no multiplicities. Then $\{e^{x \cdot z}\}_{z \in V}$ is a FAS for $\ker_{C^\infty(\Omega)} A$.*

By identifying real and imaginary parts, (Harkonen et al., 2023) used this result to construct *real* Gaussian Process (GP) priors with realizations which satisfy the PDE exactly. This method, EPGP,

successfully built probability distributions of PDE solutions from data at points, in some cases obtaining results of several orders of magnitude better than state-of-the-art PINN methods.

This paper follows this general idea of a probabilistic model for solutions of PDE systems and replaces the exponential functions $e^{x \cdot z_j}$ from EPGP by new functions which satisfy boundary conditions too.

## 3    EPGP FOR BOUNDARY VALUE PROBLEMS

We introduce and demonstrate B-EPGP on realistic problems with boundary conditions. Additional concrete examples can be found in Appendices I.1, I.2, I.3, K.1, K.3, L,M.1, M.3.

*Remark* 3.1.  Boundary conditions can be incorporated into EPGP by conditioning the GP on data at the boundary. Consider, for example, the initial-boundary value problem for the 1D heat equation:

$$\begin{cases} \partial_t u - \partial_x^2 u = 0 & \text{in } [0,1] \times [-2,2] \\ u(0,x) = f(x) & \text{for } x \in [-2,2] \\ u(t,\pm 2) = 0 & \text{for } t \in [0,1]. \end{cases}$$

A direct EPGP implementation starts with $u(t,x) = \sum_{j=1}^r w_j e^{z_j^2 t + z_j x}$ with $z_j, w_j \in \mathbb{C}$, and fits the real/imaginary parts of $u$ to the real/imaginary parts of the initial and boundary data at finite sets $X \subset [-2,2]$ and $T \subset [0,1]$. We apply this approach to the 2D wave equation in Section I.3. This method performs reasonably well in low dimensions, where boundaries can be approximated with few data points. However, in higher dimensions, the curse of dimensionality makes this approach highly inefficient, as $g^{d-1}$ points are needed for a boundary in $d$-dimensional space with a grid containing $g$ points in each dimension. This corresponds to the typical curse of dimensionality in numerical approximations. B-EPGP encodes the boundary condition directly into the FAS elements, *avoiding the curse of dimensionality* for encoding boundaries completely when encoding the boundary. Of course, like any spectral or GP method, representing fields in the interior and conditioning on data still incurs dimension-dependent costs. Furthermore, B-EPGP is not affected by the curse of dimensionality even for inhomogeneous boundary conditions, see Appendix M.

### 3.1    B-EPGP: BASES SATISFYING BOUNDARY CONDITIONS IN HALFSPACES

We begin our introduction of B-EPGP with the simplest class of boundaries: halfspaces. By a change of variables, we may assume the domain is $\Omega = \mathbb{R}_+^n := \mathbb{R}^{n-1} \times [0,\infty)$. We will later generalize this approach to more complex boundaries and provide corresponding examples.

Our goal is to construct GP priors whose realizations satisfy

$$\begin{cases} A(\partial)u = 0 & \text{in } \mathbb{R}^{n-1} \times [0,\infty) \\ B(\partial)u = 0 & \text{on } \mathbb{R}^{n-1} \times \{0\}, \end{cases} \tag{1}$$

where $B \in \mathbb{C}[\partial]^{h \times 1}$ is a linear PDE operator representing boundary conditions. Starting from Theorem 2.1, we consider realizations of the form $u(x) = \sum_{j=1}^r w_j e^{x \cdot z_j}$ with $w_j \in \mathbb{C}$ and $z_j \in V$, which we aim to modify to satisfy both PDE and boundary conditions *exactly*. In contrast, the baseline approach from Remark 3.1 enforces the boundary condition only *approximately* and at a much higher cost. We stress that we are interested in *underdetermined systems* (1), i.e., systems with many solutions. This means that our method constructs probability distributions on infinite dimensional spaces–the kernels in system (1). We only consider PDE with real coefficients, $A \in \mathbb{R}[\partial]$, $B \in \mathbb{R}[\partial]^{h \times 1}$. Thus, the gap between the complex ansatz $u$ above and constructing real GP priors is filled by taking real/imaginary parts of $u$. An example is computed in Section I.3.

For $x = (x', x_n) \in \mathbb{R}^{n-1} \times \mathbb{R}$, evaluating $B(\partial)u = 0$ at $x_n = 0$ yields $\sum_{j=1}^r w_j B(z_j) e^{x' \cdot z_j'} = 0$ for all $x' \in \mathbb{R}^{n-1}$, where $z_j'$ denotes the first $n-1$ components of $z_j$. For this identity to hold, we must have that $z_j'$ is constant with respect to $j$. Hence, $\sum_{j=1}^r w_j B(z_j) = 0$ for all $x' \in \mathbb{R}^{n-1}$. This enables us to write an algorithm to construct FAS elements for solutions of (1).

Let $z' \in \mathbb{C}^{n-1}$ be such that there exists $z_n \in \mathbb{C}$ with $(z', z_n) \in V$, and define $V'$ as the projection of the characteristic variety $V$ onto $\mathbb{C}^{n-1}$. For each $z' \in V'$, let $S_{z'} = \{z \in V : z = (z', z_n) \text{ for some } z_n \in \mathbb{C}\}$ denote the non-empty fiber over $z'$. These fibers can be computed using

Gröbner basis with negligible complexity in practice, see Appendix B. To ensure the boundary condition is satisfied, we seek weights $w_z \in \mathbb{C}$ such that

$$\sum_{z \in S_{z'}} w_z B(z) = 0. \tag{2}$$

The admissible choices of $w_z$ form the left syzygy module of the vertically stacked matrices $B(z)$ for $z \in S_{z'}$, which is again computable via Gröbner basis algorithms with negligible complexity, see again Appendix B. For each $z' \in V'$, we obtain the FAS candidate

$$\left\{ \sum_{z \in S_{z'}} w_z B(z) e^{x \cdot z} \colon \sum_{z \in S_{z'}} w_z B(z) = 0 \right\}_{z' \in V'} \tag{3}$$

of terms for the frequency $z'$. We use a finite union of such FAS elements to perform linear regression as in EPGP. The bases (3) can be computed *explicitly* by hand for the heat and wave equations with either Dirichlet or Neumann boundary conditions.

*Example* 3.2 (Dirichlet boundary condition). If we set $B = 1$, we obtain $\sum w_z = 0$ in (2). This is a broadly used condition with which we can solve explicitly. $\triangle$

*Example* 3.3. Consider the 1D heat equation $\partial_t u - \partial_x^2 u = 0$ with Dirichlet boundary condition, $B = 1$ and $u(t, 0) = 0$. Start with $z = (\zeta^2, \zeta) \in V$ and note that $z' = \zeta^2$, so that $S_{z'} = \{\pm \zeta\}$. We conclude that in this case the FAS elements are $\{e^{t\zeta^2 + x\zeta} - e^{t\zeta^2 - x\zeta}\}_{\zeta \in \mathbb{C}}$. $\triangle$

*Example* 3.4. Similarly, for the 2D wave equation $\partial_t^2 u - \partial_x^2 u - \partial_y^2 u = 0$ with Dirichlet boundary condition $u(t, 0, y) = 0$, we obtain the FAS $\{e^{\pm\sqrt{a^2+b^2}t + ax + by} - e^{\pm\sqrt{a^2+b^2}t - ax + by}\}_{a,b \in \mathbb{C}}$. $\triangle$

*Example* 3.5 (Neumann boundary condition). Another widely used boundary condition is Neumann, i.e., $B(z) = z_n$, which leads to $\sum w_z z_n = 0$. Comparing to the previous examples, this leads to the FAS $\{e^{t\zeta^2 + x\zeta} + e^{t\zeta^2 - x\zeta}\}_{\zeta \in \mathbb{C}}$ for 1D heat equation (Example 3.3) and $\{e^{\sqrt{a^2+b^2}t + ax \pm by} + e^{\sqrt{a^2+b^2}t + ax \mp by}\}_{a,b \in \mathbb{C}}$ for 2D wave equation (Example 3.4). $\triangle$

Since the calculations behind Examples 3.2-3.5 are particularly insightful, we will derive them here for Example 3.4. By Theorem 2.1, the FAS for the 2D wave equation is $e^{\pm\sqrt{a^2+b^2}t + ax + by}$ for $a, b \in \mathbb{C}$. Our boundary condition is at $x = 0$, so the projection of the frequency $z = (\sqrt{a^2 + b^2}, a, b)$ to $V'$ is $z' = (\sqrt{a^2 + b^2}, 0, b)$. The fiber $S_{z'} = \{(\sqrt{a^2 + b^2}, a, b), (\sqrt{a^2 + b^2}, -a, b)\}$, therefore our FAS candidate takes the form

$$w_+ e^{\pm\sqrt{a^2+b^2}t + ax + by} + w_- e^{\pm\sqrt{a^2+b^2}t - ax + by}.$$

This function must be 0 when $x = 0$, so we must have $w_- = -w_+$, leading to the claim in Example 3.4. The last step also follows from Eq. 2 with $B = 1$ or Example 3.2.

The calculations above can be extended to arbitrary dimensions and for arbitrary hyperplanes, see Appendix D. This new FAS thus replaces the FAS of exponential functions in EPGP. Importantly, we can prove that our method constructs approximations of all solutions to the boundary value problem:

**Theorem 3.6.** (3) *is FAS for* (1) *for heat/wave equation and Dirichlet/Neumann boundary conditions.*

We prove this in Appendix E, see Appendix F for details on the topology with respect to which we have density. Relevant computational results are in Sections 5 and Appendices I, J.

### 3.2 B-EPGP with polygonal boundaries

B-EPGP handles polygonal boundary conditions given by $k$ halfspaces $H_i$ for $1 \leq i \leq k$ as follows. We rephrase the computational steps from Section 3.1 in two parts, starting from a single EPGP FAS function $e^{\widehat{z} \cdot x}$ with frequency $\widehat{z} \in V$:

(i) Use the geometric arguments from above to collect all relevant frequency vectors in the fiber $S_{\widehat{z}'} \ni \widehat{z}$ in the context of the single halfspace.

(ii) Determine all linear combinations of FAS functions $e^{z \cdot x}$ for $z \in S_{\widehat{z}'}$ that satisfy the boundary condition, using the syzygy computation explained above.

To extend this to polygonal boundaries, we refine step (i) as follows, while keeping step (ii) unchanged:

(i') Initialize a singleton frequency set $S = \{\widehat{z}\}$. Iteratively apply the construction from (i) to $z \in S$ and a halfspace $H_i$, and add any resulting frequencies to $S$. Repeat until $S$ stabilizes, i.e., no new frequencies are introduced for any combination $z \in S$ and $H_i$, $1 \le i \le k$.

These steps in (i') do not terminate in general. We demonstrate how to still construct a FAS with non-termination in Example 3.8. In all remaining examples below, the computation terminates and the general case of termination is discussed in Appendix C in terms of reflection groups.

*Example* 3.7 (Wedge). We consider the 2D wave equation with Dirichlet boundary condition at $x = 0$ and Neumann boundary condition at $y = 0$:

$$\begin{cases} u_{tt} - u_{xx} - u_{yy} = 0 & (x, y) \in (0, \infty)^2, t > 0 \\ u(0, y, t) = 0 & y \in (0, \infty), t > 0 \\ u_y(x, 0, t) = 0 & x \in (0, \infty), t > 0 \end{cases}$$

The EPGP FAS for $u_{tt} - u_{xx} - u_{yy} = 0$ is given by $\{e^{\alpha x + \beta y + \tau t} \colon \alpha, \beta, \tau \in \mathbb{C}, \alpha^2 + \beta^2 = \tau^2\}$. Performing the algorithm from the beginning of the section leads to the FAS:

$$e^{\alpha x + \beta y + \tau t} - e^{-\alpha x + \beta y + \tau t} + e^{\alpha x - \beta y + \tau t} - e^{-\alpha x - \beta y + \tau t},$$

which satisfies both boundary conditions at once. The calculation of the FAS and an implementation with a $45°$ wedge are described in Appendices K.2, K.3, respectively. △

*Example* 3.8 (Infinite slab). Consider the 1D wave equation on a line segment $(0, \pi)$ with Dirichlet boundary at both $x = 0$ and $x = \pi$,

$$\begin{cases} u_{tt} = u_{xx} & x \in (0, \pi), t \in \mathbb{R} \\ u(0, t) = u(\pi, t) = 0 & t \in \mathbb{R}. \end{cases} \tag{4}$$

An EPGP FAS for this equation is $e^{\sqrt{-1}\xi(x \pm t)}$ for $\xi \in \mathbb{R}$, where $\sqrt{-1}$ is the imaginary unit. Section 3.1 yields the FAS $e^{\pm\sqrt{-1}\xi t} \sin(\xi x)$ that also adheres to the boundary condition at $x = 0$. Choosing $\xi \in \mathbb{Z}$ gives a FAS candidate which satisfies the condition at $x = \pi$ as well. We will prove that this candidate actually is a FAS for (4) in Section K.1, where we establish the following: △

**Theorem 3.9.** *B-EPGP gives* $\{e^{\sqrt{-1}jx} \sin(jx)\}_{j \in \mathbb{Z}}$ *as FAS of solutions of* (4).

The main reason why this holds is the classic *separation of variables method*, which extends the result at once to higher dimensions, cf. Section E:

*Example* 3.10 (Rectangle). The wave equation in a square with Dirichlet boundary conditions reads:

$$\begin{cases} u_{tt} - u_{xx} - u_{yy} = 0 & \text{for } x, y \in (0, \pi),\ t > 0 \\ u = 0 & \text{if } x \text{ or } y = 0 \text{ or } \pi. \end{cases} \tag{5}$$

In a similar fashion to Example 3.8, B-EPGP can be used to arrive at the FAS

$$e^{\pm\sqrt{-1}\sqrt{j^2 + k^2}t} \sin(kx) \sin(jy) \quad \text{for } j, k \in \mathbb{Z}. \tag{6}$$

In this case we retrieve the method of separation of variables (a classic Fourier series approach). The calculation of the FAS and an implementation are described in Appendix I.1. △

**Theorem 3.11.** *B-EPGP gives the FAS* (6) *of solutions of* (5).

In Theorem E.1 we give a general statement which incorporates both Theorems 3.9 and 3.11.

B-EPGP defines a valid GP and thus offers a fully probabilistic regression model. It not only handles noisy data but also allows sampling from underdetermined PDEs–that is, systems with non-unique solutions despite boundary conditions. In contrast, our previous examples were sufficiently constrained to yield nearly unique solutions. B-EPGP can also be conditioned on arbitrary (noisy or exact) observations. For example, consider the 2D wave equation with Dirichlet boundary conditions:

$$\begin{cases} u_{tt} - u_{xx} - u_{yy} = 0 & \text{for } x, y > 0 \\ u = 0 & \text{on } x \in \{0, 1\} \text{ and } y \in \{0, 1\} \end{cases}$$

Figure 1 shows snapshots of a random B-EPGP samples at five timepoints, illustrating its ability to capture uncertainty from sparse data.

### 3.3 Hybrid B-EPGP

For domains with both flat and curved pieces of the boundary, we use B-EPGP to fulfill the linear boundary conditions and the direct method from Remark 3.1 using data for the curved pieces.

*Example* 3.12 (Circular sector). Consider the space-time domain $\Omega = \{(x, y, t) \colon x^2 + y^2 < 1, x > 0, y > 0, t > 0\}$ which has the spatial boundary $\Gamma = ([0, 1] \times \{0\}) \cup (\{0\} \times [0, 1]) \cup \{(\cos\theta, \sin\theta) \colon \theta \in [0, \pi/2]\}$. We consider the 2D wave equation in this circular sector, $u_{tt} - u_{xx} - u_{yy} = 0$ in $\Omega$ with Dirichlet boundary condition $u = 0$ on $\Gamma$. To account for the flat pieces of $\Gamma$, B-EPGP gives a FAS similar to the one in Example 3.7, namely

$$e^{\alpha x + \beta y + \tau t} - e^{-\alpha x + \beta y + \tau t} - e^{\alpha x - \beta y + \tau t} + e^{-\alpha x - \beta y + \tau t} \quad \text{for } \alpha, \beta, \tau \in \mathbb{C} \text{ and } \alpha^2 + \beta^2 = \tau^2.$$

To account for the circular piece, we sample many points $(\cos\theta, \sin\theta, t)$ and restrict our GP to satisfy $u = 0$ at those points. For experiments, see Section 5.3. $\triangle$

### 3.4 Construction of the B-EPGP priors

We make the leap between the algebraic FAS we constructed so far and the implementation of GPs. In all examples (including the earlier contribution EPGP) we write our priors as $u(x) = \sum_{j=1}^{r} w_j b(x; z_j) = \hat{u}(x; \theta)$ where $w_j \in \mathbb{C}$ and the FAS functions $b(\,\cdot\,; z)$ are complex-valued; the variable $\theta$ aggregates all parameters $w_j, z_j$. We work with PDE and boundary conditions which have real coefficients and therefore $\hat{u}$ can also be assumed real-valued. Therefore we can write

$$\hat{u}(x; \theta) = \sum_{j=1}^{r} \Re w_j \Re b(x; z_j) - \Im w_j \Im b(x; z_j),$$

which gives a $2r$-variate Gaussian prior (assuming $(\Re w_j, \Im w_j)_{j=1}^{r} \sim \mathcal{N}(0, \Sigma)$) which also satisfies the PDE and boundary conditions. We also assume $u(x_h) - \hat{u}(x_h; \theta) \sim \mathcal{N}(0, \sigma)$ i.i.d. for each data point $x_h \in \Omega$. Regarding implementation, the number $r$ and the initial parameters for $\sigma, \Sigma, z_j$ are inserted by the user then trained by maximizing the log likelihood of the model, see also Section F. See Section I.3 for a calculation of an explicit FAS for EPGP. Any passage from *complex* FAS to *real* GP priors in our work is implemented along the same lines.

## 4 Related ML Approaches to PDEs

The intersection of machine learning and PDEs has grown into a vibrant research field with a variety of modeling paradigms. In this section, we survey classes of methods, including neural operators, GPs, and hybrid models, placing our B-EPGP framework in context.

A dominant class of approaches uses neural networks either to approximate the solution directly or as implicit function approximators. A foundational work in this domain is that of physics-informed neural networks (PINNs) (Raissi et al., 2019), which penalize deviation from the PDE operator in the loss function. Variants and improvements of this idea abound, such as variational PINNs (Kharazmi et al., 2021), Fourier PINNs (Wang et al., 2021), and gradient-enhanced PINNs (Yu et al., 2022). These approaches train neural networks to minimize the PDE and approximate data at the same time. For a similar approach to GPs, see (Chen et al., 2022).

Physics-informed neural networks with hard constraints—often called *Ansatz PINNs*—enforce boundary and initial conditions exactly by constructing a trial solution $\hat{u} = g + h\,N_\theta$, where $g$ satisfies the prescribed conditions and $h$ vanishes on the constraint sets (Lagaris et al., 1997). Recent advances make this strategy practical on complex geometries via smooth signed/approximate distance functions, $R$-functions, and transfinite interpolation, enabling exact Dirichlet enforcement and robust handling of mixed boundary types (Sukumar & Srivastava, 2022; Wang et al., 2023). Relative to penalty-based ("soft") formulations, hard-constraint designs reduce loss-weight tuning, improve optimization conditioning, and often train faster and more stably, while remaining compatible with data terms and inverse problems (Berrone et al., 2023; Barschkis, 2023). Extensions cover Neumann/Robin conditions (via tailored derivative constraints or energetic forms), high-order PDEs (first-order reformulations), and time-dependent problems (time-factorized ansätze), establishing hard-constraint PINNs as a strong default when admissible trial spaces can be constructed (Gladstone

et al., 2022; Chen et al., 2025). In contrast to these approaches, B-EPGP exactly satisfies both boundary conditions *and* the system of PDEs.

Neural operators (Kovachki et al., 2023), such as DeepONets (Lu et al., 2021) and Fourier Neural Operators (FNOs) (Li et al., 2021), represent mappings between function spaces and are trained across families of PDEs. Most neural operators face difficulties enforcing hard boundary constraints, especially in high-dimensional or complex geometries. Recent contributions have tackled multi-scale problems (Fang et al., 2023) and high-frequency regimes (Raonic et al., 2024). Neural operators are connected to GPs (Magnani et al., 2024) via linearization. The approaches (Hennig et al., 2015; Pförtner et al., 2022; Schober et al., 2014; Zhang et al., 2025) improve upon numerical PDE-solvers by casting them in a probabilistic framework.

GP models have been applied to both forward and inverse problems involving PDEs. Classical works encode linear constraints through covariance functions (Macêdo & Castro, 2008; Solin et al., 2018; Särkkä, 2011). A special case are GPs for latent force models constructed in (Alvarez et al., 2009; Hartikainen & Sarkka, 2012). Specifically, WIGPR (Henderson et al., 2021) considers the 3D wave equation. Jidling et al. (2018) sketched an approach to construct a parametrization for general linear constant coefficient PDE systems to construct Gaussian processes from them. (Lange-Hegermann, 2018) argued that this approach worked precisely for controllable PDEs and provided an algorithm to compute the parametrization using Gröbner bases. Theoretical results underlying these approaches can be found in (Henderson et al., 2023a; Sullivan, 2024) and for related approaches to neural networks see (Hendriks et al., 2020; Ranftl, 2023). All of these approaches are special cases or our approach in that they are "controllable", which means that the characteristic variety consists of all frequencies as the sole component. Boundary conditions in such controlable GPs with differential equations have been modeled via vertical scaling with polynomial (Lange-Hegermann, 2021) and analytic (Lange-Hegermann & Robertz, 2022) functions. These two papers need to modify the prior, whereas our work conditions the EPGP prior on boundary conditions. A special case was later introduced as BCGP (Dalton et al., 2024b).

Well-posed boundary values problems where only observations of the source term exist and no observations of the solution have been considered with GPs using spectral expansions of covariance functions (Gulian et al., 2022), including Lie symmetries (Dalton et al., 2024a). EPGP has recently been generalized to inverse problems (Li et al., 2025). In probabilistic numerics, (Cockayne et al., 2017) constructs meshless GP solvers that propagate discretisation uncertainty in PDE-constrained Bayesian inverse problems. Chen et al. (2021) give a rigorous collocation-based GP framework for nonlinear PDEs. Wenk et al. (2019) use GP gradient matching for fast parameter identification in nonlinear ODEs. Li et al. (2024) propose PDE-informed GPs (PIGP) that encode nonlinear PDEs via manifold constraints for parameter inference. Long et al. (2022) introduce AutoIP, a unified framework that injects general differential-equation structure into GP likelihoods. Physics-informed state-space GPs such as PHYSS-GP (Hamelijnck et al., 2024) and latent-force or probabilistic-exponential integrator models (Alvarez et al., 2009; Hartikainen & Sarkka, 2012; Bosch et al., 2023) likewise incorporate linear (and sometimes nonlinear) differential operators into GP models, typically via residual terms, state-space constructions, or problem-specific constraints. In contrast to these collocation- or likelihood-based approaches, B-EPGP directly constructs GP priors whose sample paths satisfy both a given constant-coefficient linear PDE system and its boundary conditions exactly, by combining the Ehrenpreis–Palamodov representation of the global solution space with linear boundary operators.

There is an extensive treatment of Linear PDE coming from numerics and optimization, including finite element methods (Larson & Bengzon, 2013; Logan, 2011; Johnson, 2009; Brenner, 2008; Braess, 2001), finite difference methods (Ervedoza & Zuazua, 2012; Zuazua, 2005; Langtangen & Linge, 2016; 2017; Thomas, 2013), and both (Cohen & Pernet, 2017; Mazumder, 2015; Ames, 2014; Esfandiari, 2017). We do not compare with them since they use deterministic models for well-posed problems, while we construct probabilistic models for under-determined problems.

Physics informed machine learning models play an important role in ODE theory and control. (Geist & Trimpe, 2020) constructs GP models for control for rigid body dynamics, while (Besginow & Lange-Hegermann, 2022) constructs such models for general linear ODE systems with constant coefficients, leading to control approaches (Besginow et al., 2025; Tebbe et al., 2024).

Table 1: The median $L_1$ error in $[0,4] \times [0,8]$ for the experiment from Section 5.1 shows the superiority of B-EPGP for different numbers of data points $n$.

| Algorithm | Absolute L1 Error | | Relative L1 Error | |
|:---:|:---:|:---:|:---:|:---:|
| | $n = 121$ | $n = 1201$ | $n = 121$ | $n = 1201$ |
| CNO | 7.24e-3 | 1.05e-3 | 1.31% | 0.79% |
| FNO | 1.05e-2 | 3.13e-3 | 1.95% | 1.17% |
| EPGP | 2.36e-4 | 6.62e-5 | 0.52% | 0.14% |
| B-EPGP (ours) | **1.96e-4** | **3.41e-5** | **0.37%** | **0.06%** |
| BCGP | 3.32e-4 | 6.34e-5 | 0.62% | 0.11% |
| WIGPR | 5.12e-4 | 8.34e-5 | 0.86% | 0.21% |

## 5 EXPERIMENTAL COMPARISON

B-EPGP is, as a GP, a probabilistic framework for describing the solution set of certain PDE systems. Our Theorems and their proofs show that all realizations of B-EPGP are solutions and moreover that realizations are dense in the set of solutions. To show the usefulness of these theoretical guarantees, we repurposed the B-EPGP priors as solvers by conditioning on enough data. This allows to compare B-EPGP to Neural Operators, the state-of-the-art in solving PDEs from initial values, and EPGP, the state-of-the-art in probabilistic modeling of linear PDEs with constant coefficients. Results for further examples and details can be found in Appendices I.1, I.2, I.3, J, K.1, K.3, L, M.1, M.3, including different equations and geometries, inhomogeneous linear systems, but also ablations like the necessary number of basis functions or the behavior near singularities.These include (i) details on the calculation of the B-EPGP FAS and plots for solving the wave and heat equations in several domains in App. I, K, L, (ii) comparison of our method with EPGP in App. J, (iii) an extension of our method to include inhomogeneous systems $A(\partial)u = f$ and $B(\partial)u = g$ as opposed to our homogeneous case study of (1) in App. M. We measure the accuracy of solutions in two ways: (a) verify the accuracy of our solvers when we know the *unique* solution to the PDEs, (b) check the conservation of energy, an important physical invariant, for the wave equation in bounded domains (Hansen et al., 2023). See Appendix H for details. We refrain from checking the error of a solution in the PDE or boundary condition as it unfairly favors B-EPGP, which satisfies them exactly. These experiments also demonstrate that B-EPGP can be applied in practice under various circumstances.

### 5.1 COMPARISON TO STATE OF THE ART MODELS

We compare to the Convolutional Neural Operators (CNO) (Raonic et al., 2024) and Fourier Neural Operators (FNO) (Li et al., 2021). As experimental setting we use the 1D wave equation with Neumann boundary condition,

$$\begin{cases} u_{tt} = u_{xx} & \text{in } (0, \infty) \times (0, \infty) \\ u(0, x) = h_1(x) \text{ and } u_t(0, x) = h_2(x) & \text{for } x \in [0, \infty) \\ u_x(t, 0) = 0 & \text{for } t \in (0, \infty) \end{cases}$$

where $h_1(x) = f(x-3) + f(x+3) + g(x-1) + g(x+1)$ and $h_2(x) = f'(x-3) - f'(x+3)$ for $f(x) = e^{-5x^2}$, $g(x) = e^{-10x^2}$. We compare to the unique exact solution given by

$$f(x+t-3) + f(x-t+3) + \tfrac{1}{2}\left(g(x+t-1) + g(x-t-1) + g(x+t+1) + g(x-t+1)\right)$$

Appendix D encodes the Neumann boundary into B-EPGP, leading to $\{e^{\alpha(x \pm t)} + e^{\alpha(-x \pm t)}\}_{\alpha \in \mathbb{C}}$. We consider both $n = 121$ and $n = 1201$ sample points from initial condition on the $t = 0$-interval $[0, 12]$, and for EPGP, CNO and FNO, we model the boundary at $x = 0$ by adding data at $t = 0.2\mathbb{Z}_{\geq 0}$ Table 1 shows that EPGP is between one and two orders of magnitude superior to the neural operator methods, and B-EPGP improves upon EPGP, BCGP (Dalton et al., 2024b), and WIGPR (Henderson et al., 2021) by factors of around 2. Experimental details about the neural operator computations and ours can be found in Appendix N.1.

### 5.2 HIGH-DIMENSIONAL EXAMPLE: 3D WAVE EQUATION

The 3D wave equation is computationally challenging even without boundary conditions. In Appendix G, we adapt EPGP to fit both initial displacements and velocities. Here, we use B-EPGP with

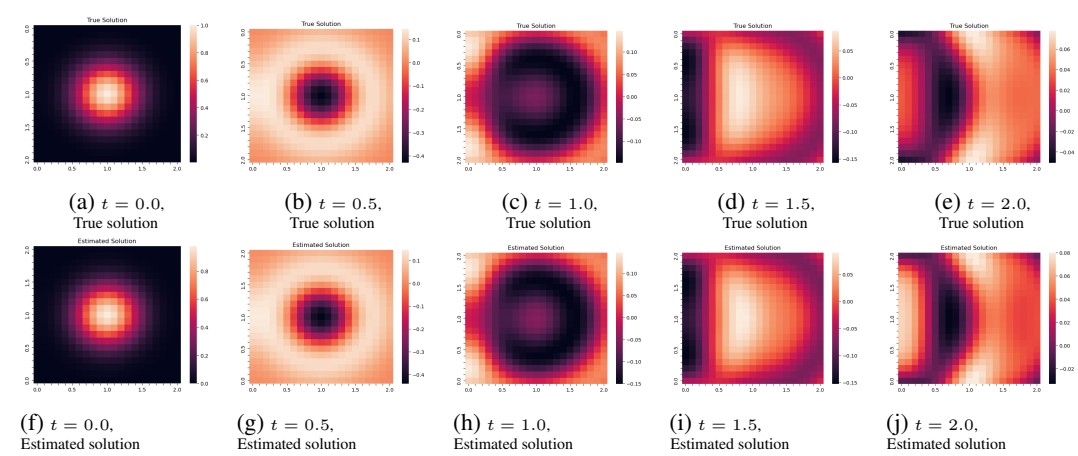

(a) $t = 0.0$,
True solution

(b) $t = 0.5$,
True solution

(c) $t = 1.0$,
True solution

(d) $t = 1.5$,
True solution

(e) $t = 2.0$,
True solution

(f) $t = 0.0$,
Estimated solution

(g) $t = 0.5$,
Estimated solution

(h) $t = 1.0$,
Estimated solution

(i) $t = 1.5$,
Estimated solution

(j) $t = 2.0$,
Estimated solution

Figure 2: Visual comparison between the true solution and B-EPGP prediction for the 3D wave equation from Section 5.2. The snapshots are taken in the plane $y = 1$.

Neumann boundary condition on the planes $y = 0$ and $z = 0$ and initial conditions as follows.

$$\begin{cases} u_{tt} = u_{xx} + u_{yy} + u_{zz} & \text{for } x \in \mathbb{R}, y, z, t > 0 \\ u = e^{-5r_1^2} + e^{-5r_2^2} + e^{-5r_3^2} \text{ and } u_t = 0 & \text{at } t = 0 \\ u_y = 0 \text{ and } u_z = 0 & \text{at respectively } y = 0 \text{ and } z = 0 \end{cases}$$

where $r_1, r_2, r_3$ denote distances between $(x, y, z)$ and $(1, 1, 1), (1, -1, 1), (1, 1, -1)$, respectively.

The B-EPGP FAS can be computed using the ideas in Section K.2 as

$$e^{ax+by+cz+dt} + e^{ax-by+cz+dt} + e^{ax+by-cz+dt} + e^{ax-by-cz+dt}$$
$$+ e^{ax+by+cz-dt} + e^{ax-by+cz-dt} + e^{ax+by-cz-dt} + e^{ax-by-cz-dt} \quad \text{for } d^2 = a^2 + b^2 + c^2.$$

Computation of solutions for PDE in 4D incurs a curse of dimensionality, hence few papers tackle 3D wave equation. For instance, EPGP runs out of memory on an Nvidia A100 80GB. Our B-EPGP finishes with a low L1-error of 0.00088. For a comparison of other computation times of machine learning models and finite element solvers, see Appendix A. These machine learning methods take training times in the range of (at least) hours and might even need training data that needs to be generated in the range of days or weeks. However, they have quick inference times (less than a second). FEM solvers are very dependent on the precise circumstances of the setup, but for the above examples usually take minutes (for limited accuracy) to hours. Training B-EPGP (determining the frequencies) takes 2371 seconds, giving a full probabilistic model instead of "just" a single solution. The training details are in Appendix N.2.

## 5.3 HYBRID B-EPGP IN A CIRCULAR SECTOR

We also implement our Hybrid B-EPGP method from Section 3.3 for the 2D wave equation in a circular sector, see Example 3.12. Let $\Omega = \{x, y > 0, x^2 + y^2 \leq 4\}, t \in (0, 4)$. The equations are:

$$\begin{cases} u_{tt} - (u_{xx} + u_{yy}) = 0 & \text{in } (0, 4) \times \Omega \\ u(0, x, y) = f(x, y) \text{ and } u_t(0, x, y) = 0 & \text{in } \Omega \\ u_n = 0 & \text{on } xy = 0 \\ u = 0 & \text{on } x^2 + y^2 = 4, \end{cases}$$

where $f(x, y) = 5 \exp(-10((x-1)^2 + (y-1)^2))$. In short, we set Dirichlet boundary conditions on the arc and Neumann boundary conditions on the wedge $xy = 0$.

These boundary conditions on the flat and curved pieces of the boundary are dealt with separately in our algorithm. To account for the Neumann boundary condition on $\{x = 0\}$ and $\{y = 0\}$ we use the

B-EPGP FAS for a wedge

$$e^{at+bx+cy} + e^{at-bx+cy} + e^{at+bx-cy} + e^{at-bx-cy} \quad \text{for } a^2 = b^2 + c^2,$$

which can be calculated using the ideas of Appendix K.2. To account for the Dirichlet boundary condition on the circular arc $\Gamma = \{x^2 + y^2 = 4, x, y > 0\}$, we assign data $u(t_h, x_h, y_h) = 0$ at many points $(t_h, x_h, y_h) \in \Gamma$. We give the training details in Appendix N.3.

Snapshots of our solution are presented in Figure 4 and the conservation of energy is demonstrated in Figure 3. We reiterate that the conservation of energy over time is equivalent to our prediction satisfying *both* the equation and the boundary condition exactly, see Appendix H.

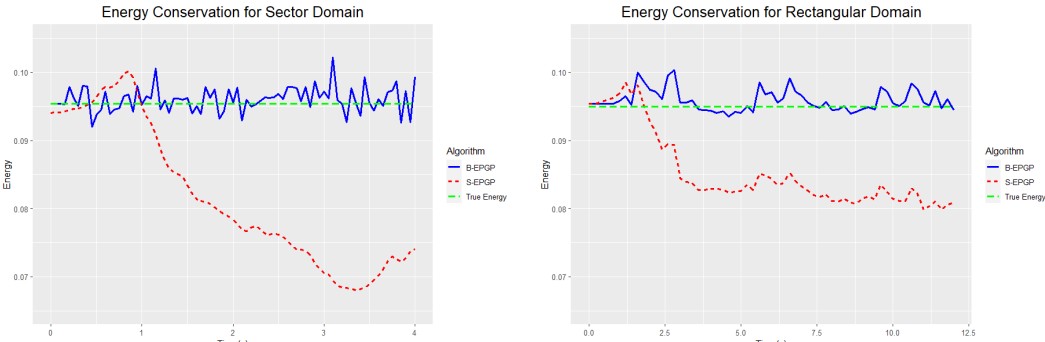

Figure 3: In Section 5.3, EPGP dissipates energy for the 2D wave equation in a sector domain, which is physically incorrect. B-EPGP only has an error due to approximating the energy integral. The same phenomenon is observed for a rectangular domain in Appendix I.1.

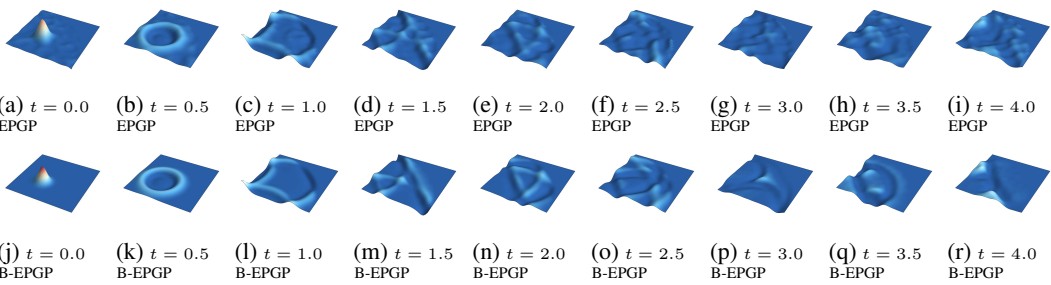

(a) $t = 0.0$ EPGP  (b) $t = 0.5$ EPGP  (c) $t = 1.0$ EPGP  (d) $t = 1.5$ EPGP  (e) $t = 2.0$ EPGP  (f) $t = 2.5$ EPGP  (g) $t = 3.0$ EPGP  (h) $t = 3.5$ EPGP  (i) $t = 4.0$ EPGP

(j) $t = 0.0$ B-EPGP  (k) $t = 0.5$ B-EPGP  (l) $t = 1.0$ B-EPGP  (m) $t = 1.5$ B-EPGP  (n) $t = 2.0$ B-EPGP  (o) $t = 2.5$ B-EPGP  (p) $t = 3.0$ B-EPGP  (q) $t = 3.5$ B-EPGP  (r) $t = 4.0$ B-EPGP

Figure 4: Solution of 2D wave equation in a sector domain calculated using the Hybrid B-EPGP method. The superiority of B-EPGP over EPGP is visually striking and is further evidenced by the energy plot in Figure 3. Animations can be found in the supplementary material as `sector_B-EPGP.mp4` and `sector_EPGP.mp4`.

## 6 SUMMARY

We introduced Boundary Ehrenpreis–Palamodov Gaussian Processes (B-EPGPs), a new framework for constructing GP priors that exactly satisfy systems of linear PDEs with constant coefficients and linear boundary conditions. Our approach generalizes existing methods by leveraging the Ehrenpreis–Palamodov theorem and extending it to include boundary constraints through symbolic algebra and Fourier analysis. Our priors are probability measures on infinite dimensional spaces, the solution sets of our underdetermined PDEs. We provided formal guarantees, explicit constructions, and demonstrated superior empirical performance over state-of-the-art neural operator models and Gaussian process models in both accuracy (by at least a factor of 2) and efficiency (by at least an order of magnitude). Our experiments are implemented in a broad class of domains which includes polygonal and curved boundaries. We do not compare with deterministic numerical solvers as our model is probabilistic.

## 7 REPRODUCIBILITY STATEMENT

We include our source code and videos of PDE solutions in the supplementary materials.

The comparison to neural operators on the 1D wave equation from Section 5.1 is described in detail in Appendix N.1. The high-dimensional example on 3D wave equation from Section 5.2 is described in detail in Appendix N.2. The hybrid B-EPGP example on the 2D wave equation that has a circular sector domain from Section 5.3 is described in detail in Appendix N.3

The free wave example from Appendix G is described in detail in Appendix N.4. The examples of 2D wave on a different bounded domain from Appendix I and Appendix K are described in detail in Appendix N.5 The comparison between B-EPGP and EPGP on the 2D wave equation from Section J is described in detail in Appendix N.6. The example of 2D heat equation from Appendix L is described in detail in Appendix N.7. The examples of inhomogeneous boundary condition from Appendix M are described in detail in Appendix N.8.

## 8 ETHICS STATEMENT

Our method constructs machine learning models constrained to physical equations. While potentially harmful use cases of our models cannot be ruled out, we do not believe that our work currently raises any questions regarding the Code of Ethics.

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

## A    STATE OF THE ART: MACHINE LEARNING FOR THE 3D WAVE EQUATION

Many recent related papers either do not tackle wave equations with boundary conditions (Henderson et al., 2023b) or do not tackle PDEs in 4D (Raonic et al., 2024). However, few papers using PINN training and neural operator learning scale to 3D wave physics (time domain and frequency domain), despite the necessity of high computational times. B-EPGP avoids this high computational times.

Neural operators fall into two dominant families. (i) *Neural operators (time domain)* learn solution *operators* for the elastic wave PDE from 3D media and source parameters to surface or volumetric wavefields. Architectures include Fourier/Factorized Fourier Neural Operators (FNO/F-FNO) and the U-shaped Neural Operator (UNO). They deliver fast surrogates once trained and generalize across media and sources (Lehmann et al., 2023; 2024). (ii) *Frequency-domain operator learning* replaces the time-domain PDE with the 3D Helmholtz equation; the Helmholtz Neural Operator (HNO) amortizes many forward solves (and even adjoint gradients) across frequencies and media with large speed/memory advantages (Zou et al., 2024).

For 3D acoustics (Helmholtz), forward PINNs reach FEM-level fields on evaluation grids (Schoder & Kraxberger, 2024; Schoder, 2025) and report wall-clock comparisons and inference times orders of magnitude below FEM once trained. These studies target low to mid frequencies in room-sized domains, and use DeepXDE/JAX/PyTorch backends with residual and boundary-loss terms.

Neural operators typically require hours to tens of hours of multi-GPU training on large synthetic corpora. Generating these synthetic corpora is also highly time intensive. After training, they evaluate in milliseconds to sub-second per case; e.g., a 3D UNO/FNO trained on 30k SEM3D simulations (surface traces) took 11 h on $4\times$A100 for 110 epochs (87M parameters) (Lehmann et al., 2023). Frequency-domain HNO reports $\sim100\times$ faster forward modeling than a spectral-element baseline and $\sim350\times$ speedup for inversion, while using $\sim40\times$ less memory than an equivalent time-domain operator (Zou et al., 2024). 3D Helmholtz PINNs report setup times of "a few hours," training 38–42.8 h, FEM baselines of 17–19 min, and PINN inference $\approx 0.05$ s ($>20,000\times$ faster than FEM at inference) (Schoder & Kraxberger, 2024).

## B    GRÖBNER BASES

Gröbner bases are a fundamental tool from computational algebra that generalize well-known algorithms in linear algebra and number theory to the setting of multivariate polynomial rings. This section provides an intuitive overview, with a focus on syzygies and how they are used in our construction of boundary-constrained Gaussian processes.

### B.1    GRÖBNER BASES

Gröbner bases extend the idea of the Gaussian elimination algorithm, which solves linear systems, to polynomial systems in several variables. Similarly, they generalize the Euclidean algorithm from univariate polynomials to multivariate settings.

Given a system of polynomial equations with multiple variables, solving or simplifying the system directly is difficult because polynomial rings in several variables are not principal ideal domains. Gröbner bases provide a canonical, algorithmically computable generating set for an ideal of polynomials that simplifies many tasks: solving systems, ideal membership testing, elimination of variables, and finding relations among polynomials.

The key idea is to choose a monomial ordering (such as lexicographic or graded reverse lexicographic order), and then reduce polynomials using a division algorithm adapted to this ordering. A Gröbner basis allows such reductions to behave predictably and mimic row operations in linear algebra.

A fully formal description of Gröbner bases exceeds the scope of this appendix. We refer to the literature for the fully story (Sturmfels, 2005; Eisenbud, 1995; Adams & Loustaunau, 1994; Greuel & Pfister, 2002; Gerdt, 2005; Buchberger, 2006). Various computer algebra systems implement Gröbner bases, most famously the open source systems Singular (Decker et al., 2024) and Macaulay2 (Grayson & Stillman, 2002).

## B.2 COMPLEXITY

Sadly, the complexity of Gröbner bases is in the vicinity of EXPSPACE completeness (cf. (Mayr, 1989; Mayr & Meyer, 1982; Bayer & Stillman, 1988; Mayr & Ritscher, 2013)). Luckily, the "average interesting example" usually terminates instantaneously (less than 0.01 seconds for all examples in this paper), and we could even compute all examples in this paper by hand.

This quick termination is to be expected. Although Gröbner-basis computation has this daunting worst-case bounds, modern algorithms (notably F4/F5) reduce it largely to structured linear algebra on Macaulay-type matrices; on many non-adversarial inputs this step is fast and rarely the overall bottleneck. F4 explicitly organizes reductions as batched Gaussian eliminations, making the cost dominated by dense linear algebra rather than term-by-term reductions (Faugère, 1999). Under generic or semi-regular assumptions, the F5 analysis predicts complexity governed by the degree of regularity, again pointing to linear-algebra–dominated runtimes (Bardet et al., 2014). In structured tasks such as computing critical points of polynomial maps, generic bounds show that the Gröbner step is tractable (e.g., $D^{O(n)}$), with experiments confirming that it typically behaves well in practice (Spaenlehauer, 2014). Worst-case phenomena remain (e.g., dimension-dependent double-exponential behavior), but these arise from carefully engineered instances rather than typical geometric or applied systems (Mayr & Ritscher, 2013). Many geometric and differential-algebraic systems carry symmetries—permutation actions, torus multigradings, or representation-theoretic structure—that dramatically shrink Gröbner computations. For permutation-invariant ideals one can work in the invariant subring or fold equivalent $S$-pairs, collapsing critical pairs and Macaulay matrices (Steidel, 2013). Equivariant Gröbner bases compute a finite basis *up to symmetry*, solving all finite quotients at once (Hillar et al.). In toric/sparse settings, Gröbner theory aligns with polyhedral combinatorics; suitable weights give squarefree initial ideals and compact universal bases (Sturmfels, 1996). Classical GL-invariant families such as determinantal ideals admit symmetry-compatible orders for which the natural generators already form Gröbner bases (Bruns & Conca, 2003).

In our case, this reduction in complexitiy holds in particular since the Gröbner basis computations only involve the comparatively small operator equations; and also larger, coupled, and more complicated PDE systems like linear elasticity or Maxwell's equations are small when compared to the (potentially big) data as inputs. In particular, also for such PDE systems, the corresponding computations terminate in fractions of a second on standard laptops.

## B.3 FIBRES

Let $V \subset \mathbb{C}^n$ be given by an ideal $I \subset \mathbb{C}[z', z_n]$ with $z' = (z_1, \ldots, z_{n-1})$, and let $\pi : V \to \mathbb{C}^{n-1}$ be the projection $(z', z_n) \mapsto z'$. Following (Barakat & Lange-Hegermann, 2022), the fibres

$$S_{z'} \;=\; \{(z', z_n) \in V\}$$

can be obtained by a single Gröbner basis computation and then inexpensive specializations. In typical use, once the Gröbner basis is computed, finding the fibre over any concrete $z'$ amounts to solving a small zero-dimensional system—fast and routine in computer algebra systems. For implementations of this and a similar algorithm we refer to the command `ConstructibleImage` in `https://homalg-project.github.io/CategoricalTowers/ZariskiFrames/` and to `https://github.com/coreysharris/TotalImage`.

## B.4 SYZYGIES

In linear algebra, a *syzygy* is simply a relation among vectors that leads to a zero vector when linearly combined, i.e. an element of the kernel of a matrix with these vectors as columns. These syzygies form a vector space.

In the more general context of polynomial systems, syzygies are relations among (vectors of) polynomials that annihilate a given combination. More precisely, given a tuple of (vectors of) polynomials $f_1, \ldots, f_m$, a syzygy is a tuple of polynomials $(g_1, \ldots, g_m)$ such that $\sum_i g_i f_i = 0$. The set of all such relations forms a module over the polynomial ring, called the *syzygy module*.

Computing a basis of this module is analogous to solving a homogeneous system of linear equations, but over a multivariate polynomial ring[1]. Instead of using the Gaussian algorithm to compute a basis, one uses Buchberger's algorithm to compute a Gröbner basis. Then, one can (in a non-trivial way called Schreyer's algorithm) read of the syzygies.

### B.5 SYZYGIES FOR B-EPGP

In our construction of *boundary-constrained Ehrenpreis–Palamodov Gaussian processes* (B-EPGP), we seek linear combinations of exponential-polynomial functions that satisfy both the differential equation and the boundary conditions exactly. These FAS functions are of the form $e^{z \cdot x}$, where $z$ lies in the *characteristic variety* associated with the PDE system.

The boundary condition introduces additional linear constraints on these exponentials. Specifically, for a fixed fiber of frequencies $S_{z'}$ (i.e., a set of vectors $z$ sharing the same spatial components), we need to find coefficients $w_z$ such that a linear combination $\sum_{z \in S_{z'}} w_z B(z)$ vanishes, where $B(z)$ represents the boundary operator evaluated at $z$.

This is precisely a syzygy computation: we seek all tuples $(w_z)_{z \in S_{z'}}$ such that a symbolic linear combination of the boundary terms gives zero. These syzygies are computed by standard Gröbner basis algorithms, and the resulting linear combinations form the building blocks for our B-EPGP FAS functions.

We comment on the role of the frequency vectors $z$ in this computation. For each fixed $S_{z'}$, the values $z \in S_{z'}$ are concrete complex numbers and the $B(z)$ are concrete complex vectors. Hence, the $w_z$ for these concrete numbers might be computed by a Gaussian algorithm. However, this induces massively redundant computations of the Gaussian algorithm for each frequency vector in each step of the training algorithm, including backpropagation through this computation. Hence, it is advantageous to symbolically precompute the $w_z$ for symbolic $z$.

For a correct computation, we need to include the polynomial relations between the $z \in S_{z'}$, i.e. that they are distinct fibers of $z'$ of the map $V \to \mathbb{C}^{n-1}$. The Gröbner basis needs to be computed over the residue class ring of the polynomial ring by these polynomial relations.

Gröbner bases and syzygies enable a symbolic guarantee that all samples from our Gaussian process model satisfy the PDE system *and* the boundary conditions *exactly*, not just approximately. This exactness is crucial in applications such as physics-informed modeling or solving ill-posed PDEs, where small violations of boundary constraints can lead to qualitatively incorrect behavior.

## C REFLECTION GROUPS AND THE TERMINATION OF THE POLYGONAL ALGORITHM

Let $\Omega$ be a polygonal domain represented as the intersection of halfspaces

$$\Omega = \bigcap_{i=1}^{m} H_i,$$

and let $V \subset \mathbb{C}^n$ denote the characteristic variety of the linear operator $A(\partial)$. The polygonal B-EPGP construction in Section 3 starts from a single EPGP frequency $\widehat{z} \in V$ and applies the halfspace construction iteratively to each boundary $H_i$.

On the level of frequencies, each halfspace $H_i$ induces a reflection

$$R_i : V \to V,$$

coming from the single-halfspace B-EPGP construction in Section 3.1. Step (i') of the polygonal algorithm can then be spelled out as follows:

- initialize a singleton frequency set

$$S_0 := \{\widehat{z}\};$$

---

[1]This is very explicitly written in the formulas (8) and (9).

- for $k = 0, 1, 2, \ldots$ define

$$S_{k+1} := S_k \cup \{R_i(z) : z \in S_k, \ 1 \le i \le m\};$$

- stop when $S_{k+1} = S_k$, and put $S := S_k$.

Conceptually, this procedure computes the orbit $S$ (using the aptly named orbit algorithm) of the initial frequency $\widehat{z}$ under the reflection group $G$ generated by the $R_i$:

$$S \ = \ G \cdot \widehat{z} \ = \ \{g\widehat{z} : g \in G\}.$$

Thus "repeat until $S$ stabilizes" simply means that we close the set of frequencies under the reflection group generated by the boundary hyperplanes. With this viewpoint, the termination issue can be stated precisely:

- If the orbit $G \cdot \widehat{z}$ is *finite*, then there exists $N$ such that $S_N = S_{N+1}$, and the iterative construction of $S$ terminates after at most $N$ steps.
- If the orbit $G \cdot \widehat{z}$ is *infinite*, then the natural spectral representation of solutions is also infinite (a Fourier series or Fourier-type expansion). In this case, one does not literally run an infinite loop, but passes to a *parametric* description of the orbit (typically by integer lattice indices).

In Section 3 we already note that the steps in (i') "do not terminate in general" and explicitly point to Example 3.8 as a canonical non-terminating case. The reflection-group interpretation above makes this dichotomy precise: termination of (i') is equivalent to finiteness of the reflection orbit $G \cdot \widehat{z}$, while an infinite orbit leads to a discrete or continuous spectral family that is handled via parameterization rather than enumeration.

### C.1 EXAMPLES WITH FINITE AND DISCRETELY PARAMETRIZED REFLECTION ORBITS

For the domains treated in this paper, the structure of the reflection orbits $G \cdot \widehat{z}$ is very simple and falls into two classes.

**Single halfspace.** For a single halfspace the group $G$ is generated by a single reflection. The orbit $G \cdot \widehat{z}$ always has size 2 (the original frequency and its reflected counterpart). This yields the familiar "even/odd" reflection formulas in Section 3.1, and the construction terminates after one step.

**Wedges with rational opening angles.** For a wedge defined by the intersection of two non-parallel halfspaces, the reflections across the two boundaries generate a dihedral reflection group. When the wedge angle is a rational multiple of $\pi$ (in particular, for a right angle), this dihedral group is finite, and so are all orbits $G \cdot \widehat{z}$. Concretely, in the $90°$ wedge of Example 3.7 and Appendix K.2, closing a single frequency under the two reflections yields a finite set of exponentials with appropriately signed combinations. The B-EPGP wedge FAS used in our experiments is obtained exactly this way, and step (i') terminates after finitely many iterations.

**Rectangles and boxes.** Rectangles (and higher-dimensional boxes) can be viewed as intersections of pairs of parallel halfspaces in each coordinate direction. Starting from a wedge, we then impose additional parallel boundaries. On the level of frequencies this does not yield a finite orbit in the naive sense (there are infinitely many reflections between two parallel boundaries), but it yields a *discrete* orbit that can be parametrized by integer indices. This is precisely what leads to the discrete set of eigenfrequencies and the resulting sine/cosine bases familiar from classical separation-of-variables; see Example 3.8 and the detailed calculations in Appendix K.

### C.2 THE SLAB AND EIGENVALUE QUANTIZATION FROM PARALLEL REFLECTIONS

The slab (or finite interval) is an instructive example precisely because the naive reflection process is infinite. We spell out here how the B-EPGP construction in Example 3.8 leads directly to the standard Fourier sine FAS, and why the word "shortcut" there refers to simplifying the algebra, not to an ad-hoc choice of FAS.

Consider the 1D wave equation on $(0, \pi)$ with Dirichlet boundary conditions at $x = 0$ and $x = \pi$,

$$\begin{cases} u_{tt} = u_{xx} & x \in (0, \pi), \ t \in \mathbb{R}, \\ u(0, t) = u(\pi, t) = 0 & t \in \mathbb{R}. \end{cases} \tag{7}$$

The construction proceeds in two steps.

**First boundary (halfspace method).** We start from a bulk exponential solution $e^{at+bx}$ with dispersion relation $a^2 = b^2$. Applying the halfspace B-EPGP construction to enforce the boundary at $x = 0$ (i.e. reflecting across $x = 0$) yields a FAS element of the form

$$e^{b(\pm t + x)} - e^{b(\pm t - x)} \ = \ 2e^{\pm bt} \sinh(bx),$$

which vanishes at $x = 0$ by construction. This is exactly step (i) applied to a single halfspace, with the reflection $b \mapsto -b$.

**Second boundary (quantization from parallel reflections).** To enforce the boundary at $x = \pi$, we require that the same expression also vanishes at $x = \pi$. This imposes the algebraic condition

$$\sinh(b\pi) \ = \ 0.$$

Thus $b$ must lie on the imaginary axis at discrete values $b \in \sqrt{-1}\,\mathbb{Z}$. Writing $b = \sqrt{-1}\,j$ with $j \in \mathbb{Z}$ and using $\sinh(\sqrt{-1}\,jx) = \sqrt{-1}\,\sin(jx)$, we obtain the FAS elements

$$e^{\pm\sqrt{-1}\,jt} \sin(jx), \qquad j \in \mathbb{Z},$$

which are exactly the usual Fourier sine modes for the wave equation on the interval. The infinite reflection orbit between two parallel boundaries manifests itself as the quantization condition $\sinh(b\pi) = 0$; solving this condition is the eigenvalue quantization step that produces the discrete set $b \in \sqrt{-1}\,\mathbb{Z}$.

In this sense, the "shortcut" in Appendix K.1 is not a manual Fourier ansatz. It is the closed-form solution of the reflection-based B-EPGP construction: we first derive the halfspace FAS, then impose the second boundary to obtain an algebraic condition, and finally solve this condition explicitly. The same pattern reappears in higher dimensions: for rectangular domains we first calculate wedge bases (finite reflection orbits), and then impose additional parallel boundaries to obtain discrete lattices of eigenfrequencies.

# D  CALCULATION OF B-EPGP FAS FOR HEAT AND WAVE EQUATIONS

## D.1  DIRICHLET AND NEUMANN CONDITIONS ON HALFSPACES

We will look at systems

$$\begin{cases} A(\partial)u = 0 & \text{in } \mathbb{R}^n_+ \\ B(\partial)u = 0 & \text{on } \mathbb{R}^{n-1} \end{cases}$$

where the boundary conditions are Dirichlet ($B = 1$) or Neumann ($B = \partial_n$). These are some of the most common boundary conditions used in PDE. As in the body of our paper, $A$ will be assumed to be a single equation and $u$ a scalar field (in Appendix F we will explain the extension to systems and vector fields). By an affine change of variable, $\mathbb{R}^n_+$ can be replaced with any other halfspace.

Inspired by the Ehrenpreis–Palamodov Theorem 2.1, we will investigate linear combinations of exponential solutions

$$u(x) = \sum_{j=1}^{M} w_j e^{x \cdot z_j}$$

which satisfy $A(\partial)u = 0$ if

$$\sum_{j=1}^{M} w_j A(z_j) e^{x \cdot z_j} = 0,$$

which implies $A(z_j) = 0$. This gives our first restriction $z_j \in V = \{z \in \mathbb{C}^n \colon A(z) = 0\}$, where $V$ is the *characteristic variety* of $A$. We proceed to incorporate the boundary condition as well. For this,

we write $x = (x', x_n)$ and in general $z'$ is the projection of $z \in \mathbb{C}^n$ on $\mathbb{C}^{n-1}$ and $z = (z', z_n)$. We get

$$\sum_{j=1}^{M} w_j B(z_j) e^{x' \cdot z'_j} = 0,$$

which now does *not* imply $B(z_j) = 0$ since some of the exponentials may be identical, i.e., some $z_j \in V$ may have the same $z'_j$. Therefore, for each unique $z'_J$, we will have that

$$\sum_{\{j \,:\, z'_j = z'_J\}} w_j B(z_j) = 0.$$

This motivates our definition of the **boundary characteristic constructible set**

$$V' = \{\zeta \in \mathbb{C}^{n-1} : \zeta = z' \text{ for some } z \in V\}.$$

Generically, this constructible set is a variety, which we then call **boundary characteristic variety**. This gives the ansatz for the B-EPGP FAS

$$\left\{ \sum_{z \in V \text{ s.t. } z' = \zeta} w_z e^{x \cdot z} : \sum_{z \in V \text{ s.t. } z' = \zeta} w_z B(z) = 0 \right\}_{\zeta \in V'}.$$

We simplify this explicitly for Dirichlet and Neumann boundary conditions, first Dirichlet:

$$\left\{ \sum_{z \in V \text{ s.t. } z' = \zeta} w_z e^{x \cdot z} : \sum_{z \in V \text{ s.t. } z' = \zeta} w_z = 0 \right\}_{\zeta \in V'}. \tag{8}$$

and then Neumann

$$\left\{ \sum_{z \in V \text{ s.t. } z' = \zeta} w_z e^{x \cdot z} : \sum_{z \in V \text{ s.t. } z' = \zeta} w_z z_n = 0 \right\}_{\zeta \in V'}. \tag{9}$$

We will calculate this explicitly for examples in the following.

### D.2 WAVE EQUATION

We will calculate the B-EPGP bases for Dirichlet and Neumann for the 2D wave equation in the domain $y > 0$. The calculations extend easily to arbitrary dimensions, but the formulas are cumbersome and we do not include them here. Our calculations extend to all halfspaces parallel to the $t$-axis by affine changes of variable. Considering boundary conditions on halfspaces that are not parallel to the time axis, which would violate the physical meaning of initial and boundary conditions.

We start with Dirichlet conditions. To be specific we look at

$$\begin{cases} u_{tt} - (u_{xx} + u_{yy}) = 0 & \text{in } \{(t, x, y) : y > 0\} \\ u = 0 & \text{at } (t, x, 0). \end{cases}$$

In this case,

$$V = \{(a, b, c) \in \mathbb{C}^3 : a^2 = b^2 + c^2\},$$

so $a = \pm\sqrt{b^2 + c^2}$, meaning $a$ can be any complex root of $z^2 - (b^2 + c^2) = 0$. In this case it we have that

$$V' = \mathbb{C}^2,$$

since for any $(a, b) \in \mathbb{C}^2$ there is at least one (and generically two) $c \in \mathbb{C}$ such that $(a, b, c) \in V$.

We proceed with computing the FAS for Dirichlet boundary conditions, so we substitute in (8) to get that for any $a, b \in \mathbb{C}$, the vectors $(a, b, c) \in V$ are given by $c = \pm\sqrt{a^2 - b^2}$, so if we write $z^{\pm} = (a, b, \pm\sqrt{a^2 - b^2})$, the relations in (8) are

$$w_{z^+} + w_{z^-} = 0 \implies w_{z^-} = -w_{z^+},$$

which leads to the FAS

$$\{e^{at+bx+\sqrt{a^2-b^2}y} - e^{at+bx-\sqrt{a^2-b^2}y}\}_{a,b\in\mathbb{C}}.$$

This can be rearranged as

$$\{e^{\pm\sqrt{a^2+b^2}t+ax+by} - e^{\pm\sqrt{a^2+b^2}t+ax-by}\}_{a,b\in\mathbb{C}},$$

which is what we have in Example 3.4.

In the case of Neumann boundary conditions, the calculation is similar. Using the same notation for $z^{\pm}$ as above, we get from (9) that

$$cw_{z^+} - cw_{z^-} = 0 \implies w_{z^-} = w_{z^+},$$

which leads to the FAS

$$\{e^{at+bx+\sqrt{a^2-b^2}y} + e^{at+bx-\sqrt{a^2-b^2}y}\}_{a,b\in\mathbb{C}}.$$

This can be rearranged as

$$\{e^{\pm\sqrt{a^2+b^2}t+ax+by} + e^{\pm\sqrt{a^2+b^2}t+ax-by}\}_{a,b\in\mathbb{C}},$$

which is what we have in Example 3.5.

### D.3 HEAT EQUATION

We will proceed similarly to the previous subsection. The same considerations concerning higher dimensions and choosing various halfspaces apply. We look at

$$\begin{cases} u_t - (u_{xx} + u_{yy}) = 0 & \text{in } \{(t,x,y): y > 0\} \\ u = 0 & \text{at } (t,x,0). \end{cases}$$

In this case,

$$V = \{(a,b,c) \in \mathbb{C}^3 : a = b^2 + c^2\},$$

so $a = b^2 + c^2$. In this case it we have that

$$V' = \mathbb{C}^2,$$

since for any $(a,b) \in \mathbb{C}^2$ there is at least one (and generically two) $c \in \mathbb{C}$ such that $(a,b,c) \in V$.

We proceed with computing the FAS for Dirichlet boundary conditions, so we substitute in (8) to get that for any $a$, $b \in \mathbb{C}$, the vectors $(a,b,c) \in V$ are given by $c = \pm\sqrt{a - b^2}$, so if we write $z^{\pm} = (a, b, \pm\sqrt{a - b^2})$, the relations in (8) are

$$w_{z^+} + w_{z^-} = 0 \implies w_{z^-} = -w_{z^+},$$

which leads to the FAS

$$\{e^{at+bx+\sqrt{a-b^2}y} - e^{at+bx-\sqrt{a-b^2}y}\}_{a,b\in\mathbb{C}}.$$

This can be rearranged as

$$\{e^{(a^2+b^2)t+ax+by} - e^{(a^2+b^2)t+ax-by}\}_{a,b\in\mathbb{C}},$$

which is slightly more general than what we have in Example 3.3.

In the case of Neumann boundary conditions, the calculation is similar. Using the same notation for $z^{\pm}$ as above, we get from (9) that

$$cw_{z^+} - cw_{z^-} = 0 \implies w_{z^-} = w_{z^+},$$

which leads to the FAS

$$\{e^{at+bx+\sqrt{a-b^2}y} + e^{at+bx-\sqrt{a-b^2}y}\}_{a,b\in\mathbb{C}}.$$

This can be rearranged as

$$\{e^{(a^2+b^2)t+ax+by} + e^{(a^2+b^2)t+ax-by}\}_{a,b\in\mathbb{C}},$$

which is slightly more general than what we have in Example 3.5 for the 1D heat equation.

# E  PROOF THAT B-EPGP GIVES ALL SOLUTIONS FOR HEAT AND WAVE EQUATIONS

We will prove Theorem 3.6 which applies to heat $\partial_t u - \Delta u = 0$ and wave equations $\partial_t^2 u - \Delta u = 0$ (here we use the notation $\Delta u = \partial_{x_1}^2 + \partial_{x_2}^2 + \ldots + \partial_{x_n}^2$ for the Laplacian operator) and any halfspace parallel to the time axis. Since the Laplacian operator is invariant under rotations, we can perform an affine change of variable to reduce the boundary condition to the plane $x_1 = 0$. Our proof below stems from the fact that, in the case of both equations, given a solution in $\{x_1 > 0\}$ with Dirichlet (resp. Neumann) boundary conditions on $\{x_1 = 0\}$, its odd (resp. even) extension with respect to $\{x_1 = 0\}$ will satisfy the same equation in full space.

We will only show the calculations in the case of the 2D wave equation with Dirichlet boundary condition. Increasing the dimension barely changes the argument. The modification required to deal with the heat equation is also minimal (one less integration by parts in time). To deal with the Neumann boundary condition, one uses even extension instead of odd extension in the calculation.

*Proof of Theorem 3.6.* Writing $\Box u = u_{tt} - u_{xx} - u_{yy}$, we consider smooth solutions of

$$\begin{cases} \Box u = 0 & \text{for } x > 0, y \in \mathbb{R}, t > 0 \\ u(t, 0, y) = 0 & \text{for } y \in \mathbb{R}, t > 0. \end{cases}$$

The main observation is that the odd extension (even for Neumann boundary condition) of $u$ is a solution in full space of the wave equation. Let

$$v(t, x, y) = \begin{cases} u(t, x, y) & \text{for } x \geq 0 \\ -u(t, -x, y) & \text{for } x < 0. \end{cases}$$

Clearly $\Box v(t, x, y) = 0$ for $x \neq 0$. We still need to check that $\Box v = 0$ across $x = 0$, but since the odd extension need not have two classical derivatives, we compute the *distributional derivative* across $x = 0$. To see this let $\phi \in C_c^\infty(\{t > 0\})$ be a test function. We write $dV = dxdydt$ and integrate by parts

$$\int v\Box\phi dV = \int_{x>0} u\Box\phi dV - \int_{x<0} u(t, -x, y)\Box\phi(t, x, y)dV$$

$$= -\int_{x=0} u\phi_x dydt - \int_{x>0} u_t\phi_t - u_x\phi_x - u_y\phi_y dV - \int_{x=0} u\phi_x dydt$$

$$+ \int_{x<0} u_t(t, -x, y)\phi_t(t, x, y) + u_x(t, -x, y)\phi_x(t, x, y) - u_y(t, -x, y)\phi_y(t, x, y)dV$$

$$= -\int_{x=0} u_x\phi dydt + \int_{x>0} \phi\Box u dV + \int_{x=0} u_x\phi dydt + \int_{x<0} \phi\Box u dV = 0,$$

so indeed $\Box v = 0$ in full space. Since $v$ is odd, we have

$$v(t, x, y) = \tfrac{1}{2}v(t, x, y) - \tfrac{1}{2}v(y, -x, y),$$

so for $x > 0$ we obtain

$$u(t, x, y) = \tfrac{1}{2}u(t, x, y) - \tfrac{1}{2}u(y, -x, y). \tag{10}$$

By Ehrenpreis–Palamodov Theorem 2.1 we know that we can approximate

$$u(t, x, y) \approx \sum_j c_j e^{\alpha_j t + \beta_j x + \gamma_j y}$$

for some $\alpha_j, \beta_j, \gamma_j \in \mathbb{C}$ with $\alpha_j^2 = \beta_j^2 + \gamma_j^2$. By (10), it follows that

$$u(t, x, y) \approx \sum_j \tfrac{1}{2}c_j(e^{\alpha_j t + \beta_j x + \gamma_j y} - e^{\alpha_j t - \beta_j x + \gamma_j y}).$$

This is exactly the FAS produced by our B-EPGP algorithm, see Example 3.4. □

We next look at the heat and wave equation in rectangles and give a generalization of Theorems 3.9 and 3.11:

**Theorem E.1** (Heat and Wave in rectangles). *Let $A = \partial_t - \Delta$ or $A = \partial_{tt}^2 u - \Delta u$, $\Omega = [0, L_1] \times [0, L_2] \times \ldots \times [0, L_n]$, and $B$ be Dirichlet or Neumann boundary condition on $\partial\Omega \times \mathbb{R}$. Consider the equation*

$$\begin{cases} Au = 0 & \text{in } \Omega \times \mathbb{R} \\ Bu = 0 & \text{on } \partial\Omega \times \mathbb{R}. \end{cases}$$

*Then B-EPGP gives the FAS*

$$\{e^{\sqrt{-1}ht} f(j_1 x_1 \pi/L_1) f(j_2 x_2 \pi/L_2) \ldots f(j_n x_n \pi/L_n) \colon j_1, j_2, \ldots, j_n \in \mathbb{Z}\},$$

*where*

- $h = j_1^2 + j_2^2 + \ldots + j_n^2$ *for heat equation and* $h = \pm\sqrt{j_1^2 + j_2^2 + \ldots + j_n^2}$ *for wave equation,*

- $f = \sin$ *for Dirichlet boundary condition and* $f = \cos$ *for Neumann boundary condition.*

*Proof.* We will only cover the case of the wave equation with Dirichlet boundary conditions. We will make some simplifications which do not restrict the idea of proof: we set $n = 2$ so $u = u(t, x, y)$ and $L_1 = L_2 = \ldots = L_n = \pi$.

Let $u$ be a solution of

$$\begin{cases} \Box u = 0 & \text{for } x \in (0, \pi), y \in (0, \pi) \\ u = 0 & \text{for } x \text{ or } y = 0 \text{ or } \pi. \end{cases}$$

We define the extension to $x, y \in (0, 2\pi)$ by

$$v(t, x, y) = \begin{cases} u(t, x, y) & x \in (0, \pi), y \in (0, \pi) \\ -u(t, x - \pi, y) & x \in (\pi, 2\pi), y \in (0, \pi) \\ -u(t, x, y - \pi) & x \in (0, \pi), y \in (\pi, 2\pi) \\ u(t, x - \pi, y - \pi) & x \in (\pi, 2\pi), y \in (\pi, 2\pi). \end{cases}$$

We extend $v$ by periodically in $(x, y)$ with cell $(0, 2\pi)^2$ to $\mathbb{R}^{1+2}$ without changing its name. We make several observations:

- For each $t$, the function $(x, y) \mapsto v(t, x, y)$ is $(0, 2\pi)^2$-periodic, therefore $v$ has a Fourier expansion $\sum_{j,k\in\mathbb{Z}} f_{j,k}(t) e^{i(jx+ky)}$.

- For each $(t, x)$, the function $y \mapsto v(t, x, y)$ is odd, therefore only the terms $\sin(ky)$ will appear in the Fourier expansion.

- Similarly $x \mapsto v(t, x, y)$ is odd, so $v$ has a Fourier expansion $\sum_{j,k\in\mathbb{Z}} f_{j,k}(t) \sin(jx) \sin(ky)$.

- $v$ solves the equation in full space, $\Box v = 0$ in $\mathbb{R}^{1+2}$.

Only the last assertion is non-obvious and requires a careful distributional calculation as in the proof of Theorem 3.6 above, taking $\phi \in C_c^\infty(\mathbb{R}^{1+2})$ and showing that $\int u\Box\phi dV = 0$ by careful integration by parts. We omit the details.

Finally, we plug in $v = \sum_{j,k\in\mathbb{Z}} f_{j,k}(t) \sin(jx) \sin(ky)$ in the equation $\Box v = 0$ to obtain $f_{j,k}''(t) + (j^2 + k^2) f_{j,k}(t) = 0$, which is an ODE with linearly independent solutions $e^{\pm\sqrt{-1}\sqrt{j^2+k^2}t}$. This gives us the Fourier FAS

$$e^{\pm\sqrt{-1}\sqrt{j^2+k^2}t} \sin(jx) \sin(ky) \quad \text{for } j, k \in \mathbb{Z},$$

which is the FAS computed using B-EPGP in Example 3.10. This coincides with the *separation of variables* method. $\qquad\square$

The same extension works for the heat equation with Dirichlet boundary condition. For the Neumann boundary condition (for both equations), one removes the two "minus" signs in rows 2 and 3 of the definition of $v$; its periodic extension is thus an even function. Higher dimensions take more effort to set up $v$ on $2^n$ branches, but the idea is the same.

## F EHRENPREIS–PALAMODOV THEOREM AND EPGP

We will begin with a very precise version of the statement that *exponential-polynomial solutions are dense in the space of all solutions* for a linear PDE system. For notational clarity, we write $\mathbb{C}[\partial] := \mathbb{C}[\partial_1, \ldots, \partial_n]$, $x = (x_1, \ldots, x_n)$, and $z = (z_1, \ldots, z_n)$.

**Theorem F.1.** *Let $\Omega \subset \mathbb{R}^n$ be an open convex set. Let $A = A(\partial) \in \mathbb{C}[\partial]^{\ell \times k}$ be an operator matrix of a system of linear PDEs with constant coefficients and $V = \{z \in \mathbb{C}^n \colon \ker A(z) \neq 0\}$ its characteristic variety. Then there exist a decomposition $V = \bigcup_{i=1}^m V_i$ into irreducible varieties $V_i$ and a set of vector polynomials in $2n$ variables called* Noetherian multipliers $\{p_{i,j}\}_{j=1\ldots r_i, i=1,\ldots m} \subset \mathbb{C}[x,z]^k$ *such that solutions of the form*

$$\sum_{h=1}^N \sum_{i=1}^m \sum_{j=1}^{r_i} c_{i,j,h} p_{i,j}(x, z_{i,j,h}) e^{x \cdot z_{i,j,h}} \quad \text{with } z_{i,j,h} \in V_i \qquad \text{(EP)}$$

*are dense in the space of smooth solutions of $A(\partial)u(x) = 0$ in $\Omega$.*

Next, we clarify the notion of smooth solution and the topology with respect to which we have density. First, we write $\mathcal{F} = C^\infty(\Omega)$ to be the space of smooth functions $\Omega \to \mathbb{C}$. This is a Frechét space under the standard topology induced by the semi-norms $s_{a,b}(u) = \max_{|\alpha|=a, x \in \Omega_b} |\partial^\alpha u(x)|$, where $\Omega_b \uparrow \Omega$ is an increasing sequence of compact sets which exhausts $\Omega$. This is to say that a sequence $u_q \to u$ in $\mathcal{F}$ if $s_{a,b}(u_q - u) \to 0$ for all $a, b$. Algebraically, $\mathcal{F}^k$ is a $\mathbb{C}[\partial]$-module under the action of differentiation.

Our solution space is then

$$\ker_\mathcal{F} A = \{u \in \mathcal{F}^k \colon A(\partial)u = 0\}.$$

Ehrenpreis–Palamodov Theorem states that each element $u \in \ker_\mathcal{F} A$ can be approximated by $u_q \to u$ in $\mathcal{F}^k$ with solutions $u_q$ of the form (EP).

The algorithm EPGP from (Harkonen et al., 2023) revolves around fitting coefficients $c_{i,j,h}$ and "frequencies" $z_{i,j,h}$ in formula (EP). To simplify notation, we will simply write $\{b(x; z)\}_{z \in V}$ for the continuously indexed FAS that we are working with (exponential-polynomial solutions of $A(\partial)u = 0$). Predictions are written in the form

$$\phi(x) = \sum_{j=1}^N c_j b(x; z_j). \qquad (11)$$

We will assume that our solution is given as data points $y_h \approx u(x_h)$ for $h = 1, \ldots M$, where $N \ll M$. We write $C = (c_j) \in \mathbb{C}^N$, $Z = (z_j) \in V^N$, $X = (x_h) \in \mathbb{R}^M$, $Y = (y_h) \in \mathbb{C}^M$.

We will model $C \sim \mathcal{N}(0, \Sigma)$ as multivariate Gaussian distribution with covariance $\Sigma = \text{diag}(\sigma_j^2)_{j=1}^N$. We will also assume that the data has Gaussian noise, so $Y - \phi(X) \sim \mathcal{N}(0, \sigma_0^2 I_M)$. Writing $B = (b(x_h; z_j)) \in \mathbb{C}^{M \times N}$, we have that $\phi(X) = BC$, so $\phi(X) \sim \mathcal{N}(0, B\Sigma B^T)$. We write $\sigma^2 = (\sigma_j^2)_{j=0}^N$ for the vector of parameters of the underlined distributions. The marginal log likelihood for this model is maximized if the function

$$L(Z, \sigma^2; X, Y) = \frac{1}{\sigma_0^2}(|Y|^2 - Y^H B A^{-1} B^H Y) + (M - N) \log \sigma_0^2 + \log \det \Sigma + \log \det A,$$

is minimized, where $A = N\sigma_0^2 \Sigma^{-1} + B^H B$ and $H$ denotes the conjugate-transpose operation. We then use stochastic gradient descent to minimize $L(Z, \sigma^2)$. Once we obtain $Z$, we plug in the explicit formula for $C = A^{-1} B^H Y$ and use (11) as our prediction.

We use the same regression model in the present paper, by choosing $b(x; z)$ according to the B-EPGP FAS instead of the EPGP FAS.

## G    FREE WAVE EQUATION IN 2D AND 3D

In fact, our first improvement of EPGP is to update it to include initial velocity as well as initial displacement. We will explain this for the example of the wave equation in arbitrary dimension $n$.

Even without boundary conditions, this is an important example for which ongoing research is being developed (Henderson et al., 2023b). Writing $\Box u = u_{tt} - u_{x_1 x_1} - u_{x_2 x_2} - \ldots - u_{x_n x_n}$, the problem to consider is the **Free Wave Equation**:

$$\begin{cases} \Box u = 0 & \text{in } \mathbb{R}^n \times (0, \infty) \\ u(0, x) = f(x) & \text{for } x \in \mathbb{R}^n \\ u_t(0, x) = g(x) & \text{for } x \in \mathbb{R}^n . \end{cases}$$

Vanilla EPGP only deals with the case $g = 0$. Our main observation is that if $u$ is a GP solution with covariance kernel $k$, then $(u, u_t)$ is a GP with covariance kernel

$$\begin{bmatrix} k(x, t; x', t') & \partial_t k(x, t; x', t') \\ \partial_{t'} k(x, t; x', t') & \partial^2_{tt'} k(x, t; x', t'), \end{bmatrix}$$

which we fit to data $(u(0, X), u_t(0, X)) = (f(X), g(X))$. Mathematically, this is the same as considering the PDE system

$$\begin{cases} \Box u = 0 & \text{in } \mathbb{R}^n \times (0, \infty) \\ v - u_t = 0 & \text{in } \mathbb{R}^n \times (0, \infty) \\ u(0, x) = f(x) & \text{for } x \in \mathbb{R}^n \\ v(0, x) = g(x) & \text{for } x \in \mathbb{R}^n . \end{cases}$$

This can then be solved using Vanilla EPGP for $A(u, v) = (\Box u, v - u_t)$ with initial condition for both $u$ and $v$.

As an example, we will consider the 2D case,

$$\begin{cases} u_{tt} = u_{xx} + u_{yy} & \text{in } \mathbb{R}^2 \times (0, \infty) \\ u(0, x, y) = f(x - 2) + f(y - 2) & \text{in } \mathbb{R}^2 \\ u_t(0, x, y) = f'(x - 2) + f'(y - 2) & \text{in } \mathbb{R}^2, \end{cases}$$

where $f(x) = \exp(-5x^2)$ or $f(x) = \cos(5x)$. The true solution is given by $u(t, x, y) = f(x + t - 2) + f(y + t - 2)$ and our results can be found in Figure 5.

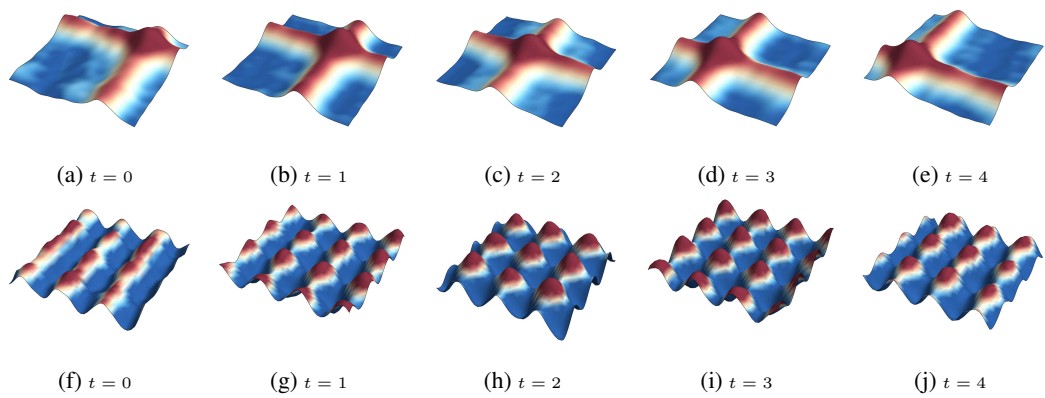

(a) $t = 0$    (b) $t = 1$    (c) $t = 2$    (d) $t = 3$    (e) $t = 4$

(f) $t = 0$    (g) $t = 1$    (h) $t = 2$    (i) $t = 3$    (j) $t = 4$

Figure 5: Solutions to the initial boundary value problem for the 2D wave equation (no boundary conditions). Solution fits both initial condition and initial speed. This is a necessary improvement of the EPGP algorithm from (Harkonen et al., 2023), which can produce non-physical solutions in certain cases. Animations can be found in the supplementary material as `free_velocity1.mp4` and `free_velocity2.mp4`.

# H  CHECK USING CONSERVATION OF ENERGY

So far, we checked the accuracy of our results either by comparison with true solutions (wherever we could construct them) or by comparison with other solvers. For certain equations, there is another mathematical tool that we can use, namely conservation of energy:

**Theorem H.1.** *Let $\Omega \subset \mathbb{R}^d$ be a convex, bounded, open set and consider a smooth solution $u \in C^\infty(\bar{\Omega} \times [0, \infty))$ of the wave equation*

$$\Box u = 0 \quad in \; \Omega \times (0, \infty).$$

*Suppose that $u$ satisfies Dirichlet or Neumann boundary conditions*

$$u = 0 \quad or \quad \frac{\partial u}{\partial n} u = 0 \quad for \; x \in \partial\Omega.$$

*Then the **energy** of the solution $u$*

$$E(t) = \int_\Omega \left| \frac{\partial u(x,t)}{\partial t} \right|^2 + \sum_{j=1}^d \int_\Omega \left| \frac{\partial u(x,t)}{\partial x_j} \right|^2 dx$$

*is constant for all $t \geq 0$.*

*Proof.* We can see that

$$E(t) = \int_\Omega u_t^2 + |\nabla u|^2 dx$$

Then we can integrate by parts to obtain

$$\begin{aligned} E'(t) &= \int_\Omega [2u_t u_{tt} + 2(\nabla u \cdot \nabla u_t)] \, dx \\ &= \int_\Omega 2u_t u_{tt} dx - 2 \int_\Omega (\Delta u) u_t dx + 2 \int_{\partial\Omega} u_t \frac{\partial u}{\partial n} dS \\ &= \int_\Omega 2u_t(u_{tt} - \Delta u) dx + 2 \int_{\partial\Omega} u_t \frac{\partial u}{\partial n} dS \\ &= 2 \int_{\partial\Omega} u_t \frac{\partial u}{\partial n} dS. \end{aligned}$$

For either Dirichlet or Neumann boundary condition, we have that

$$\int_{\partial\Omega} u_t \frac{\partial u}{\partial n} dS = 0,$$

Therefore, $E(t)$ is indeed constant. $\qquad \square$

In practice, we approximate $E(t)$ a Riemann sum of step-size $h$. For instance, when $d = 2$ we take

$$Q(t) = \frac{\text{Area}(\Omega)}{\#(h\mathbb{Z})^2 \cap \Omega} \sum_{(x,y) \in (h\mathbb{Z})^2 \cap \Omega} (u_t^2 + u_x^2 + u_y^2)\big|_{(x,y,t)}.$$

The quantity we implement in our code is

$$\tilde{Q}(hT) = \sum_{(x,y) \in (h\mathbb{Z})^2 \cap \Omega} (u_t^2 + u_x^2 + u_y^2)\big|_{(x,y,hT)} \tag{12}$$

where $T$ is the final time ($t \in [0, T]$) and we choose $h = 0.1$.

## I   WAVE EQUATION IN BOUNDED DOMAINS

In this section we will provide numerical results for the 2-D wave equation in bounded domains, meaning that we investigate $u_{tt} - (u_{xx} + u_{yy}) = 0$ in $\Omega \times (0, T)$ with Dirichlet or Neumann boundary conditions on $\partial\Omega$, where $\Omega \subset \mathbb{R}^2$ will is a bounded open convex set. We will use both EPGP (Remark 3.1) and B-EPGP methods and compare them.

We will check the validity of our results using the conservation of energy principle from Theorem H.1. We will show that by using EPGP a non-negligible amount of energy is lost/dissipated. In fact, we can say more: We will solve initial boundary value problems with given initial condition and zero initial speed, meaning that we will solve

$$\begin{cases} u_{tt} - (u_{xx} + u_{yy}) = 0 & \text{in } \Omega \times (0, T) \\ u(0, x, y) = f(x, y) & \text{at } t = 0 \\ u_t(0, x, y) = 0 & \text{at } t = 0 \end{cases}$$

which is a well-posed problem under Dirichlet or Neumann boundary conditions. By Theorem H.1 we have

$$E(t) = E(0) = \int_\Omega u_t^2(0, x, y) + u_x^2(0, x, y) + u_y^2(0, x, y) dx dy = \int_\Omega f_x^2 + f_y^2 dx dy.$$

Thus in our experiments we will compare the energy of the B-EPGP with the true value computed from initial conditions above and also show its superiority over EPGP.

### I.1   RECTANGULAR DOMAINS

We first consider $\Omega = (0, 4)^2$ and $t \in (0, 12)$ and look for the solution of the initial boundary value problem

$$\begin{cases} u_{tt} - (u_{xx} + u_{yy}) = 0 & \text{for } x, y \in (0, 4),\ t \in (0, 12) \\ u(0, x, y) = \exp(-10((x-1)^2 + (y-1)^2)) & \text{for } x, y \in (0, 4) \\ u_t(0, x, y) = 0 & \text{for } x, y \in (0, 4) \\ u(t, x, y) = 0 & \text{for } x \text{ or } y = 0 \text{ or } 4. \end{cases}$$

We will give a shortcut to our B-EPGP algorithm for finding a FAS. We consider $H_1 = \{x = 0\}$, $H_2 = \{y = 0\}$, $H_3 = \{x = 4\}$, $H_4 = \{y = 4\}$ and let $e^{at+bx+cy}$ be a solution of the wave equation, meaning that $a^2 = b^2 + c^2$. We can calculate a FAS for the wedge $\{x, y > 0\}$, see Appendix K.2, which is

$$e^{at+bx+cy} - e^{at-bx+cy} - e^{at+bx-cy} + e^{at-bx-cy}$$
$$= e^{at}(e^{bx+cy} - e^{-bx+cy} - e^{bx-cy} + e^{-bx-cy})$$
$$= e^{at}(e^{cy}(e^{bx} - e^{-bx}) - e^{-cy}(e^{bx} - e^{-bx}))$$
$$= 2\sqrt{-1}e^{at}\sin(bx)(e^{cy} - e^{-cy})$$
$$= -4e^{at}\sin(bx)\sin(cy)$$

Here we observe a shortcut: if we set $b, c \in \mathbb{Z}$, we obtain a set of functions

$$\{e^{\pm \frac{\pi\sqrt{-1}}{4}\sqrt{j^2+k^2}t}\sin(\tfrac{\pi}{4}jx)\sin(\tfrac{\pi}{4}ky)\}_{j,k\in\mathbb{Z}}$$

which satisfy the boundary condition on $H_2$ as well (see also Appendix K.1). That set of functions has linear span which is dense in the set of all solutions to (13) by Theorem E.1.

Snapshots of our solution are presented in Figure 6 and the conservation of energy is demonstrated in Figure 7.

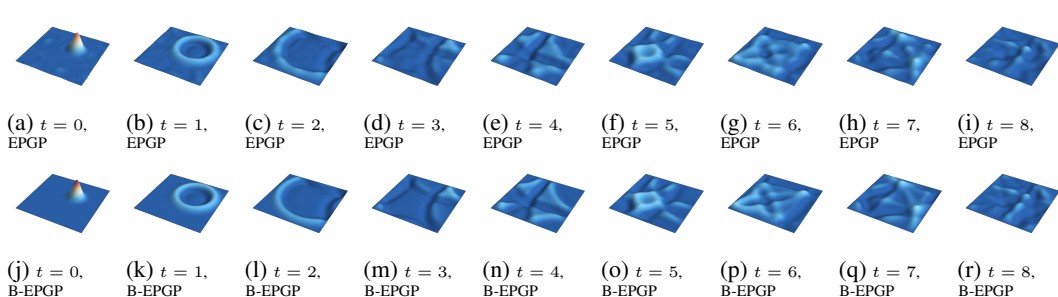

(a) $t = 0$, EPGP (b) $t = 1$, EPGP (c) $t = 2$, EPGP (d) $t = 3$, EPGP (e) $t = 4$, EPGP (f) $t = 5$, EPGP (g) $t = 6$, EPGP (h) $t = 7$, EPGP (i) $t = 8$, EPGP

(j) $t = 0$, B-EPGP (k) $t = 1$, B-EPGP (l) $t = 2$, B-EPGP (m) $t = 3$, B-EPGP (n) $t = 4$, B-EPGP (o) $t = 5$, B-EPGP (p) $t = 6$, B-EPGP (q) $t = 7$, B-EPGP (r) $t = 8$, B-EPGP

Figure 6: Solution of 2D wave equation in a rectangular domain calculated using the EPGP and B-EPGP methods. We use a Dirichlet boundary condition which is visible above from the fact that the edges of our plots have the same color in all snapshots. The results of B-EPGP seem more reasonable, as they are more stable and lack corrupting high frequencies. Animations can be found in the supplementary material as `rectangle_EPGP.mp4` and `rectangle_B-EPGP.mp4`.

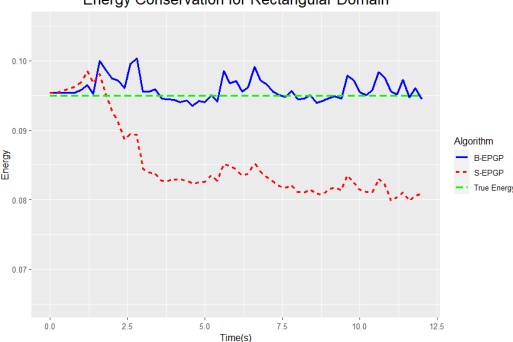

Figure 7: This figure shows the energy conservation for the 2D wave equation in a rectangular domain. We expect a constant value from physical principles. EPGP method incurs a non-negligible loss of energy. The error in B-EPGP is partly due to our approximation of the energy integral in (12).

## I.2 TRIANGULAR DOMAINS

We next consider $\Omega = \{(x, y) \colon 0 < y < x < 4\}$ and $t \in (0, 4)$ and look for the solution of the initial boundary value problem

$$\begin{cases} u_{tt} - (u_{xx} + u_{yy}) = 0 & \text{in } \Omega \times (0, 4) \\ u(0, x, y) = \exp(-10((x - 3)^2 + (y - 3)^2)) & \text{in } \Omega \\ u_t(0, x, y) = 0 & \text{in } \Omega \\ u(t, x, y) = 0 & \text{for } (x, y) \in \partial\Omega \end{cases}$$

We will use the B-EPGP FAS computed in Section I.1 and an odd reflection in the diagonal line $y = x$ which gives

$$\left\{ e^{\pm \frac{\pi\sqrt{-1}}{4}\sqrt{j^2 + k^2} t} \left[ \sin(\tfrac{\pi}{4} jx) \sin(\tfrac{\pi}{4} ky) - \sin(\tfrac{\pi}{4} kx) \sin(\tfrac{\pi}{4} jy) \right] \right\}_{j,k \in \mathbb{Z}}.$$

It is easy to see that this new FAS satisfies the Dirichlet boundary condition on all three boundary hyperplanes $\{x = 4\}, \{y = 0\}, \{x = y\}$. Snapshots of our solution are presented in Figure 8 and the conservation of energy is demonstrated in Figure 9.

## I.3 CIRCLE: DRUM MEMBRANE

Our methods extend to non polygonal domains. Here we consider the 2D wave equation in a disc with Dirichlet boundary conditions, a classic model for circular drum membranes. We will consider

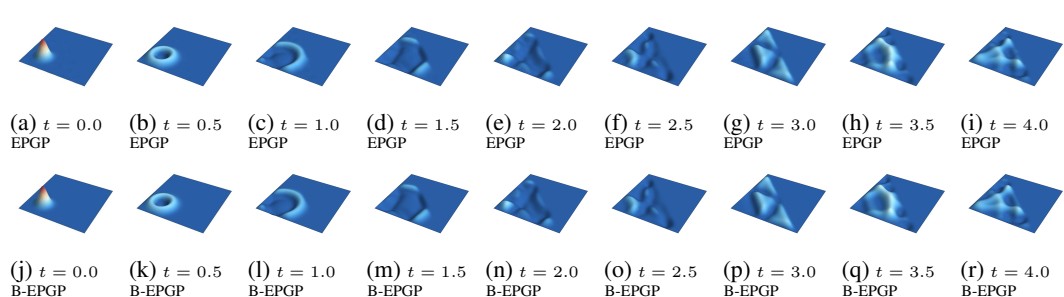

(a) $t = 0.0$ EPGP   (b) $t = 0.5$ EPGP   (c) $t = 1.0$ EPGP   (d) $t = 1.5$ EPGP   (e) $t = 2.0$ EPGP   (f) $t = 2.5$ EPGP   (g) $t = 3.0$ EPGP   (h) $t = 3.5$ EPGP   (i) $t = 4.0$ EPGP

(j) $t = 0.0$ B-EPGP   (k) $t = 0.5$ B-EPGP   (l) $t = 1.0$ B-EPGP   (m) $t = 1.5$ B-EPGP   (n) $t = 2.0$ B-EPGP   (o) $t = 2.5$ B-EPGP   (p) $t = 3.0$ B-EPGP   (q) $t = 3.5$ B-EPGP   (r) $t = 4.0$ B-EPGP

Figure 8: Solution of 2D wave equation in a triangular domain with Dirichlet boundary conditions calculated using the EPGP and B-EPGP methods. Animations can be found in the supplementary material as `triangle_EPGP.mp4` and `triangle_B-EPGP.mp4`.

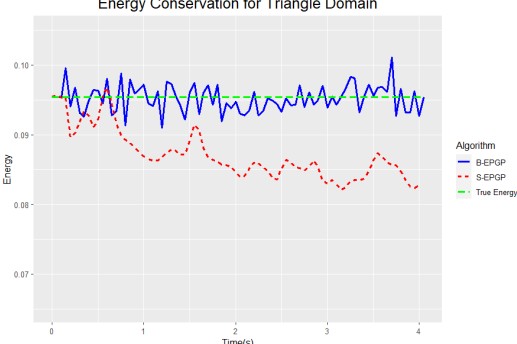

Figure 9: This figure shows the of energy conservation for the 2D wave equation in a triangular domain. We expect a constant value from physical principles. S-EPGP method incurs a non-negligible loss of energy. The error in B-EPGP is partly due to our approximation of the energy integral in (12).

the spatial domain $\Omega = \{(x, y) : x^2 + y^2 \leq 16\}$ and $t \in (0, 7)$, and the equations

$$\begin{cases} u_{tt} - (u_{xx} + u_{yy}) = 0 & \text{in } (0, 7) \times \Omega \\ u(0, x, y) = \exp(-10(x^2 + y^2)) & \text{in } \Omega \\ u_t(0, x, y) = 0 & \text{in } \Omega \\ u(t, x, y) = 0 & \text{on } (0, 7) \times \partial\Omega. \end{cases}$$

Here we cannot use B-EPGP for polygons or Hybrid B-EPGP since no piece of the boundary of $\Omega$ is polygonal. Instead, we will use the following implementation of the baseline method EPGP as described in Remark 3.1: We consider the EPGP FAS for the 2D wave equation without boundary conditions $\{e^{\pm\sqrt{a^2+b^2}t+ax+by}\}_{a,b\in\mathbb{C}}$ and model the Dirichlet boundary condition as data $u(x_h, y_h, t_h) = 0$ for points $(x_h, y_h, t_h) \in \partial\Omega \times (0, 7)$. We also give data points to represent the initial conditions at $t = 0$ as in the rest of the paper.

We clarify the formulation of the EPGP implementation, since the FAS is expressed as *complex* variable and we are using *real* Gaussian Processes. We observe that $u$ defined by the system is real-valued so $u = \Re u$. Second, due to the nature of the wave equation, we can simply take $a, b \in \sqrt{-1}\,\mathbb{R}$, where $\sqrt{-1}$ is the imaginary unit. Therefore, we start with $v_j, w_j \in \mathbb{C}$ and $a_j, b_j, c_j, d_j \in \mathbb{R}$ and

$$u(x, y, t) = \sum_{j=1}^{M} w_j \exp\left(\sqrt{-1}\left(t\sqrt{a_j^2 + b_j^2} + xa_j + yb_j\right)\right)$$

$$+ v_j \exp\left(\sqrt{-1}\left(-t\sqrt{c_j^2 + d_j^2} + xc_j + yd_j\right)\right)$$

so that since $u = \Re u$ everywhere, we obtain

$$u(x, y, t) = \sum_{j=1}^{M} \Re \left( w_j \exp \left( \sqrt{-1} \left( t\sqrt{a_j^2 + b_j^2} + xa_j + yb_j \right) \right) \right)$$

$$+ \Re \left( v_j \exp \left( \sqrt{-1} \left( -t\sqrt{c_j^2 + d_j^2} + xc_j + yd_j \right) \right) \right)$$

$$= \sum_{j=1}^{M} \Re w_j \cos \left( t\sqrt{a_j^2 + b_j^2} + xa_j + yb_j \right) - \Im w_j \sin \left( t\sqrt{a_j^2 + b_j^2} + xa_j + yb_j \right)$$

$$+ \Re v_j \cos \left( -t\sqrt{c_j^2 + d_j^2} + xc_j + yd_j \right) - \Im v_j \sin \left( -t\sqrt{c_j^2 + d_j^2} + xc_j + yd_j \right),$$

where $\Re$ and $\Im$ denote the real and imaginary part respectively. In the EPGP implementation, the coefficients $(\Re w_j, \Im w_j, \Re v_j, \Im v_j)_j = 1$ are viewed as a multivariate Gaussian distribution with $4M \times 4M$ covariance matrix which is trained, alongside the real parameters $a_j, b_j, c_j, d_j$.

Snapshots of our solution are presented in Figure 10 and the conservation of energy is demonstrated in Figure 11.

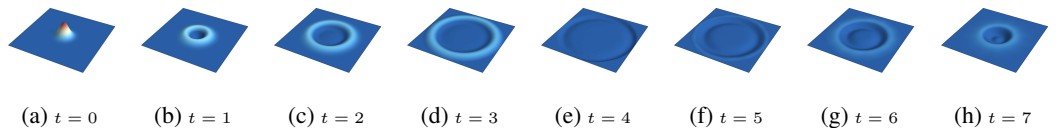

(a) $t = 0$    (b) $t = 1$    (c) $t = 2$    (d) $t = 3$    (e) $t = 4$    (f) $t = 5$    (g) $t = 6$    (h) $t = 7$

Figure 10: Radially symmetric solution to 2D wave equation in a circular domain with Dirichlet boundary conditions evaluated at 8 timepoints. The animation can be found in the supplementary material as `circle_EPGP.mp4`.

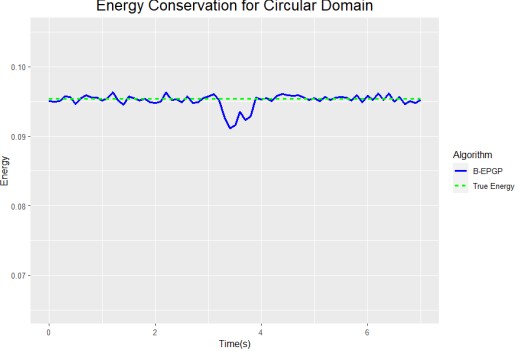

Figure 11: This figure shows the energy conservation for the 2D wave equation in a circular domain. We expect a constant value from physical principles. The error is partly due to our numerical approximation of the energy integral in (12).

## J  B-EPGP AND EPGP ON 2D WAVE EQUATION

We compare B-EPGP and EPGP in-depth, discussing accuracy and computational resources. Consider the following initial boundary value problem for the 2D wave equation in a halfspace $x > 0$:

$$\begin{cases} u_{tt} = u_{xx} + u_{yy} & \text{for } t \in (0, \infty), x \in (0, \infty), y \in \mathbb{R} \\ u(0, x, y) = f(x, y) \text{ and } u_t(0, x, y) = 0 & \text{for } x \in [0, \infty), y \in \mathbb{R} \\ u_x(t, 0, y) = 0 & \text{for } t \in (0, \infty), y \in \mathbb{R} \end{cases}$$

with initial condition $f = f_1 + f_2 + f_3$ for $c_i = -5i^2 + 20i - 10$ and

$$f_i(x, y) = J_0(c_i \cdot \sqrt{((x - i)^2 + (y - i)^2)}) + J_0(c_i \cdot \sqrt{((x + i)^2 + (y - i)^2)}),$$

Table 2: The results from Section J show the superiority of B-EPGP over EPGP when comparing the median L1-difference to the *exact* solution and the computation time. We report the standard deviation of ten repetitions.

| Algorithm | Abs Err($10^{-4}$) | Rel Err(%) | Time(s) |
|---|---|---|---|
| EPGP | $4.26 \pm 0.21$ | $1.82 \pm 0.09$ | 6059 |
| B-EPGP (ours) | $\mathbf{0.48} \pm 0.02$ | $\mathbf{0.72} \pm 0.03$ | 804 |

where $J_0$ is the Bessel function of order 0. The unique exact solution is given by

$$u(t, x, y) = f_1(x, y) \cos(5t) + f_2(x, y) \cos(10t) + f_3(x, y) \cos(5t).$$

We can use B-EPGP to obtain the FAS

$$e^{\alpha x + \beta y + \tau t} + e^{-\alpha x + \beta y + \tau t} + e^{\alpha x + \beta y - \tau t} + e^{-\alpha x + \beta y - \tau t} \text{ for } \alpha, \beta, \tau \in \mathbb{C} \text{ and } \alpha^2 + \beta^2 = \tau^2.$$

We fix the number of frequencies to be the same in both EPGP and B-EPGP. Hence, B-EPGP requires only one-quarter as many trainable parameters as EPGP. Both methods model initial data on a spatial grid of spacing 0.2. EPGP requires additional boundary data sampled on a grid, which triples the number of data and increases the size of the covariance matrix by a factor of nine, leading to substantially higher computational cost. A one-sided paired Wilcoxon signed-rank test over ten repetitions, with the alternative hypothesis that the location shift is less than zero, yields a highly significant $p$-value of $0.001 \ll 0.05$. Table 2 sums up the comparison, where B-EPGP reduces prediction error, runtime, and memory usage by an order of magnitude. Figure 13 describes how EPGP and B-EPGP perform with different data size and FAS elements. A visual comparison of the two solutions can be seen in Figure 12.

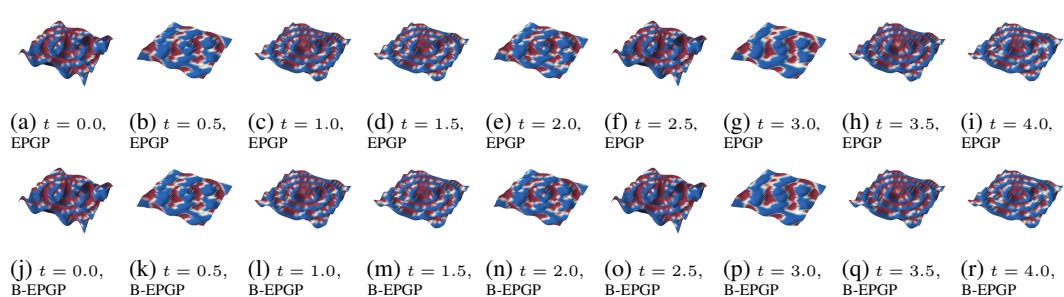

(a) $t = 0.0$, EPGP   (b) $t = 0.5$, EPGP   (c) $t = 1.0$, EPGP   (d) $t = 1.5$, EPGP   (e) $t = 2.0$, EPGP   (f) $t = 2.5$, EPGP   (g) $t = 3.0$, EPGP   (h) $t = 3.5$, EPGP   (i) $t = 4.0$, EPGP

(j) $t = 0.0$, B-EPGP   (k) $t = 0.5$, B-EPGP   (l) $t = 1.0$, B-EPGP   (m) $t = 1.5$, B-EPGP   (n) $t = 2.0$, B-EPGP   (o) $t = 2.5$, B-EPGP   (p) $t = 3.0$, B-EPGP   (q) $t = 3.5$, B-EPGP   (r) $t = 4.0$, B-EPGP

Figure 12: Predictions to the 2D wave equation from Section J using EPGP and B-EPGP. We use this experiment as a benchmark as we can compare both predictions with a highly nontrivial *exact* solution. Animations can be found in the supplementary material as `halfplane_EPGP.mp4` and `halfplane_B-EPGP.mp4`.

## K   B-EPGP BASES FOR WAVE EQUATION IN INTERSECTIONS OF TWO HALFSPACES

We will also analyze the next simplest case which is more general than halfspaces (Appendix D), namely intersections of two halfspaces. These, have only two relative positions, parallel or not, in which case we will distinguish between the case when the angle is acute or not.

### K.1   PARALLEL HALFSPACES: SLABS

Consider the equation in Example 3.8, namely 1D wave equation in an interval:

$$\begin{cases} u_{tt} = u_{xx} & x \in (0, \pi), t \in \mathbb{R} \\ u(0, t) = u(\pi, t) = 0 & t \in \mathbb{R}. \end{cases} \tag{13}$$

We will show how to use our B-EPGP algorithm to calculate a FAS. We consider $H_1 = \{x = 0\}$, $H_2 = \{x = \pi\}$ and let $e^{at+bx}$ be a solution of the wave equation, meaning that $a^2 = b^2$, so we will simply write $a = \pm b$. We obtain $e^{b(\pm t+x)}$. Using Section D, this FAS is extended to a FAS which satisfies the $H_1$ condition by

$$e^{b(\pm t+x)} - e^{b(\pm t-x)} = e^{\pm bt}(e^{bx} - e^{-bx}) = 2e^{\pm bt}\sinh(bx).$$

Here we observe a shortcut: if we set $b \in \sqrt{-1}\mathbb{Z}$, we obtain a set of functions $\{e^{\pm\sqrt{-1}jt}\sin(jx)\}_{j\in\mathbb{Z}}$ which satisfy the boundary condition on $H_2$ as well. This set has linear span which is dense in the set of all solutions to (13) by Theorem E.1.

A similar calculation gives the FAS $\{e^{\pm\sqrt{-1}jt}\cos(jx)\}_{j\in\mathbb{Z}}$ in the case of Neumann boundary conditions $u_x = 0$.

We present our solution to the initial boundary value problem (13) computed using the B-EPGP FAS above in Figure 14.

### K.2 Large wedges

We will consider the case when the two halfspaces make an angle of $90°$ (see Example 3.7) and calculate the FAS for

$$\begin{cases} \Box u(x,y,t) = 0 & (x,y) \in (0,\infty)^2 \\ u(0,y,t) = 0 & y \in (0,\infty) \\ u_y(x,0,t) = 0 & x \in (0,\infty). \end{cases}$$

We will calculate our **B-EPGP algorithm** and show the calculations in some detail.

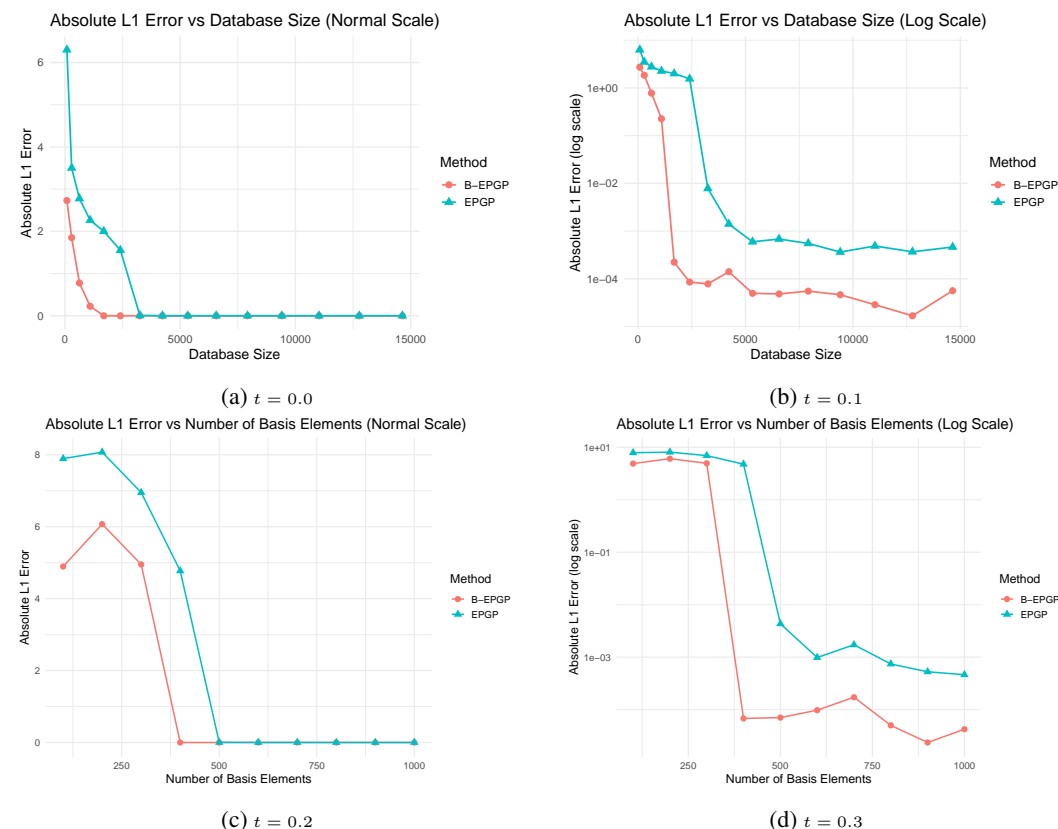

(a) $t = 0.0$

(b) $t = 0.1$

(c) $t = 0.2$

(d) $t = 0.3$

Figure 13: The relationship between the error and the data size / number of FAS elements we use, for both EPGP and B-EPGP, and both in normal scale and log scale.

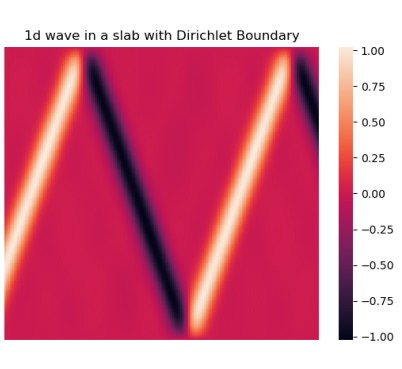
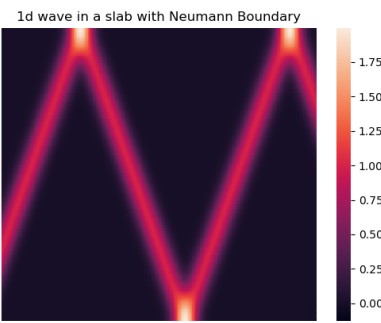

(a) Dirichlet boundary condition          (b) Neumann boundary condition

Figure 14: B-EPGP solution for 1D wave in an infinite slab with different boundary conditions. The difference color is explained by the different reflections: For Dirichlet, a negative wave is reflected ("hard boundary") and for Neumann a positive wave is reflected ("soft boundary"). This behavior is already readable from the bases calculated in Appendix K.1.

We notice that the boundary hyperplanes are $H_1 = \{x = 0\}$ and $H_2 = \{y = 0\}$.

In Step (i), we begin with one exponential solution

$$e^{at+bx+cy} \quad \text{with } a^2 = b^2 + c^2.$$

This is extended to a FAS that satisfies the $H_1$ condition by

$$e^{at+bx+cy} - e^{at-bx+cy}.$$

This follows from the calculations in Section D.2. Similarly, we extend to a FAS that satisfies the $H_2$ condition

$$e^{at+bx+cy} + e^{at+bx-cy}.$$

This gives us the intermediate "FAS", at the end of Step (ii),

$$\{e^{at+bx+cy} - e^{at-bx+cy}, e^{at+bx+cy} + e^{at+bx-cy} : a^2 + b^2 = c^2\}.$$

We proceed with Step (i') and check the both boundary conditions. We notice that each type of FAS element satisfies exactly one of the two boundary conditions, so we must return to Step (ii).

To extend the term $e^{at+bx+cy} - e^{at-bx+cy}$ (which satisfies the $H_1$ condition) to satisfy the $H_2$ condition, we use the same calculation as above to obtain

$$e^{at+bx+cy} - e^{at-bx+cy} + (e^{at+bx-cy} - e^{at-bx-cy}).$$

To extend the term $e^{at+bx+cy} + e^{at+bx-cy}$ (which satisfies the $H_2$ condition) to satisfy the $H_1$ condition, we get

$$e^{at+bx+cy} + e^{at+bx-cy} - (e^{at-bx+cy} + e^{at-bx-cy}).$$

Coincidentally, both FAS elements constructed above equal

$$e^{at+bx+cy} - e^{at-bx+cy} + e^{at+bx-cy} - e^{at-bx-cy} \quad \text{for } a^2 = b^2 + c^2, \tag{14}$$

which can easily be seen to satisfy both boundary conditions. In particular, we obtain that the algorithm terminates. Thus we obtained the FAS claimed in Example 3.7.

## K.3 SMALL WEDGES

We will also consider and also implement the case of an acute wedge, e.g. $\Omega = \{(x, y) : x > 0, y < x\}$ and $t \in (0, 8)$. We will look at the 2D wave equation with Neumann boundary conditions

$$\begin{cases} u_{tt} - (u_{xx} + u_{yy}) = 0 & \text{in } \Omega \times (0, 8) \\ u(0, x, y) = \exp(-10((x-3)^2 + (y-1)^2)) & \text{in } \Omega \\ u_t(0, x, y) = 0 & \text{in } \Omega \\ u_n(t, x, y) = 0 & \text{on } \partial\Omega \times (0, 8). \end{cases}$$

The B-EPGP FAS can be computed along the same lines as the case of the right angle above. However, the calculations are much more ample so we only state the result here:

$$e^{z_1 x + z_2 y + \tau t} + e^{-z_1 x + z_2 y + \tau t} + e^{z_1 x - z_2 y + \tau t} + e^{-z_1 x - z_2 y + \tau t}$$
$$+ e^{z_2 x + z_1 y + \tau t} + e^{-z_2 x + z_1 y + \tau t} + e^{z_2 x - z_1 y + \tau t} + e^{-z_2 x - z_1 y + \tau t}$$
$$+ e^{z_1 x + z_2 y - \tau t} + e^{-z_1 x + z_2 y - \tau t} + e^{z_1 x - z_2 y - \tau t} + e^{-z_1 x - z_2 y - \tau t}$$
$$+ e^{z_2 x + z_1 y - \tau t} + e^{-z_2 x + z_1 y - \tau t} + e^{z_2 x - z_1 y - \tau t} + e^{-z_2 x - z_1 y - \tau t} \quad \text{for } \tau^2 = z_1^2 + z_2^2.$$

We present our solution to the initial boundary value problem computed using the B-EPGP FAS above in Figure 15.

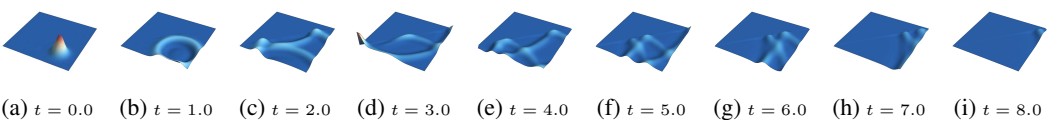

(a) $t = 0.0$   (b) $t = 1.0$   (c) $t = 2.0$   (d) $t = 3.0$   (e) $t = 4.0$   (f) $t = 5.0$   (g) $t = 6.0$   (h) $t = 7.0$   (i) $t = 8.0$

Figure 15: Solution of 2D wave equation in a $45°$ wedge domain evaluated at 9 timepoints. Since the domain is unbounded, the wave leaves the domain in finite time. The animation can be found in the suplementary material as `wedge_B-EPGP.mp4`.

## L  HEAT EQUATION IN 2D

The examples we gave so far focused on wave equations, often in 2 space dimensions as these produce the most visually striking videos and are better represented in the paper as snapshots at various times. Our method extends equally well to equations for heat, which we will give an example of in this

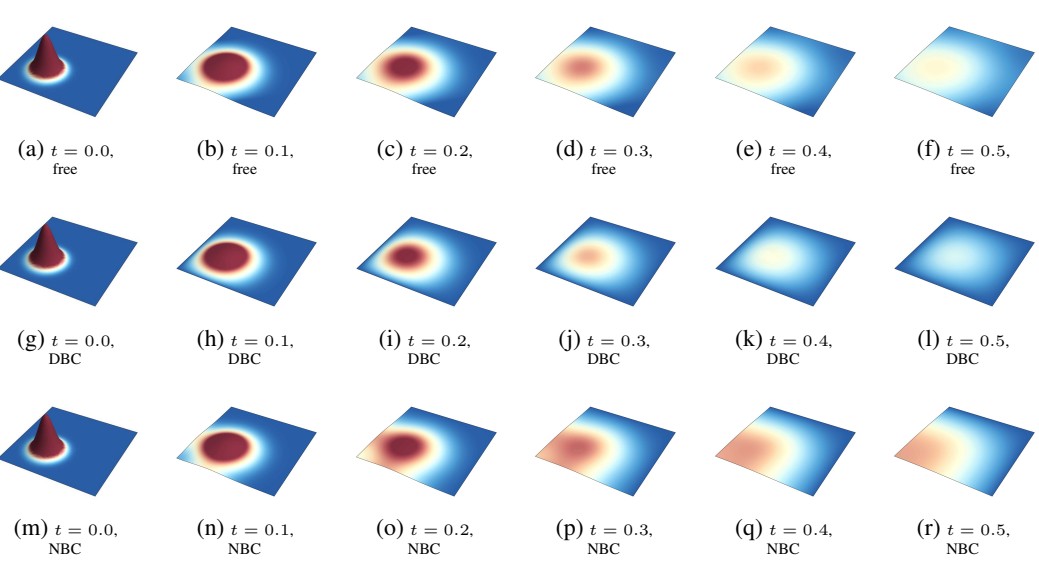

(a) $t = 0.0$, free   (b) $t = 0.1$, free   (c) $t = 0.2$, free   (d) $t = 0.3$, free   (e) $t = 0.4$, free   (f) $t = 0.5$, free

(g) $t = 0.0$, DBC   (h) $t = 0.1$, DBC   (i) $t = 0.2$, DBC   (j) $t = 0.3$, DBC   (k) $t = 0.4$, DBC   (l) $t = 0.5$, DBC

(m) $t = 0.0$, NBC   (n) $t = 0.1$, NBC   (o) $t = 0.2$, NBC   (p) $t = 0.3$, NBC   (q) $t = 0.4$, NBC   (r) $t = 0.5$, NBC

Figure 16: Snapshots of our solution for 2D heat equation with different boundary condition at 6 timepoints. The effects of the boundary conditions are very clearly visible. Row 1, free heat equation (no BC): heat diffuses exponentially in time. Row 2, Dirichlet BC: heat is maintained at 0 on the edges $x = 0$ and $y = 0$ so heat is diffused faster. Row 3, Neumann BC: the edges $x = 0$ and $y = 0$ are thermally insulated, so no heat is diffused there; hence heat is diffused slower, in directions of large $x, y$. Animations can be found in the supplementary material as `heat_free.mp4`, `heat_DBC.mp4`, and `heat_NBC.mp4`.

section. We will consider a wedge domain and Neumann boundary conditions:

$$\begin{cases} u_t - (u_{xx} + u_{yy}) = 0 & \text{in } (0,4) \times (0,\infty)^2 \\ u(0,x,y) = 5\exp(-10((x-1)^2 + (y-1)^2)) & \text{in } (0,\infty)^2 \\ u_x(t,0,y) = 0 & \text{for } t \in (0,4),\ y \in (0,\infty) \\ u_y(t,x,0) = 0 & \text{for } t \in (0,4),\ x \in (0,\infty). \end{cases}$$

In this case, we can use the calculations in Section D.3 and K.2 to obtain the B-EPGP FAS

$$e^{(a^2+b^2)t+ax+by} + e^{(a^2+b^2)t-ax+by} + e^{(a^2+b^2)t+ax-by} + e^{(a^2+b^2)t-ax-by} \quad \text{for } a,b \in \mathbb{C}.$$

For comparison, we will also consider the case of Dirichlet boundary conditions

$$\begin{cases} u_t - (u_{xx} + u_{yy}) = 0 & \text{in } (0,4) \times (0,\infty)^2 \\ u(0,x,y) = 5\exp(-10((x-1)^2 + (y-1)^2)) & \text{in } (0,\infty)^2 \\ u(t,0,y) = 0 & \text{for } t \in (0,4),\ y \in (0,\infty) \\ u(t,x,0) = 0 & \text{for } t \in (0,4),\ x \in (0,\infty), \end{cases}$$

for which we obtain the B-EPGP FAS

$$e^{(a^2+b^2)t+ax+by} - e^{(a^2+b^2)t-ax+by} - e^{(a^2+b^2)t+ax-by} + e^{(a^2+b^2)t-ax-by} \quad \text{for } a,b \in \mathbb{C}.$$

In Figure 16 we will further compare these results visually with the solution of the heat equation in full space

$$\begin{cases} u_t - (u_{xx} + u_{yy}) = 0 & \text{in } (0,4) \times \mathbb{R}^2 \\ u(0,x,y) = 5\exp(-10((x-1)^2 + (y-1)^2)) & \text{in } \mathbb{R}^2. \end{cases}$$

## M    INHOMOGENEOUS SYSTEMS

Our method is easily extended to cover the case of inhomogeneous systems,

$$\begin{cases} A(\partial)u = f & \text{in } \mathbb{R}^{n-1} \times [0,\infty) \\ B(\partial)u = g & \text{on } \mathbb{R}^{n-1} \times \{0\}, \end{cases} \tag{15}$$

where $f$ and $g$ are given functions. We first make the observation that the difference between the affine space in (15) and the linear space in (1) is only that of a particular solution $u_p$ of (15):

$$\{u\colon A(\partial)u = f, B(\partial)u = g\} = u_p + \{u\colon A(\partial)u = 0, B(\partial)u = 0\}.$$

This is a very classic observation, almost as old as the study of partial differential equations. In particular, given a particular solution $u_p$, defining probability measures on the space of solutions to the inhomogeneous system (15) is equivalent to defining probability measures on the space of solutions to the homogeneous system (1). The latter is the main achievement of B-EPGP.

This of course leaves open the issue of finding particular solutions $u_p$, which is not trivial, but can be dealt with various methods, including analytical methods and numerical solvers. We outline a possible method here, which has a spectral flavor. Write $x = (x', x_n)$, where $x' \in \mathbb{R}^{n-1}$ and $x_n \in \mathbb{R}$. For simplicity of notation, we only consider the case of single equations, i.e., $A \in \mathbb{R}[\partial]$, $B \in \mathbb{R}[\partial']$; the case of systems can be dealt with using pseudoinverses instead of division in the algebraic calculation below. We expand $f$ and $g$ in exponential series,

$$f(x) \approx \sum_{j=1}^{M} w_j e^{\sqrt{-1}z_j \cdot x}, \qquad g(x') \approx \sum_{j=1}^{M} \tilde{w}_j e^{\sqrt{-1}y_j \cdot x'},$$

with $z_j \in \mathbb{R}^n$ and $y_j \in \mathbb{R}^{n-1}$. This can be done either with deterministic or probabilistic methods for Fourier transforms.

We first set

$$u_1(x) = \sum_{j=1}^{M} A(\sqrt{-1}z_j)^{-1} w_j e^{\sqrt{-1}z_j \cdot x},$$

which solves $A(\partial)u_1 = f$ with no regard to the boundary conditions. We would next need to solve

$$\begin{cases} A(\partial)u = 0 & \text{in } \mathbb{R}^{n-1} \times [0, \infty) \\ B(\partial)u = g - B(\partial)u_1 & \text{on } \mathbb{R}^{n-1} \times \{0\}. \end{cases}$$

We write out explicitly

$$g - B(\partial)u_1 = \sum_{j=1}^{M} \tilde{w}_j e^{\sqrt{-1}y_j \cdot x'} - B(\sqrt{-1}z_j')A(\sqrt{-1}z_j)^{-1}w_j e^{\sqrt{-1}z_j' \cdot x'}$$

We then find $\tilde{y}_j, \tilde{z}_j \in \mathbb{C}$ such that $A(\tilde{y}_j) = 0 = A(\tilde{z}_j)$ and $\tilde{y}_j' = \sqrt{-1}y_j$ and $\tilde{z}_j' = z_j'$; this is an algebraic computation which can be performed in Macaulay2. Then set

$$u_2(x) = \sum_{j=1}^{M} B(\sqrt{-1}\tilde{y}_j)^{-1}\tilde{w}_j e^{\sqrt{-1}\tilde{y}_j \cdot x} - A(\sqrt{-1}\tilde{z}_j)^{-1}w_j e^{\sqrt{-1}\tilde{z}_j \cdot x}$$

to obtain a solution to the system above. Therefore $u_p = u_1 + u_2$ will solve (15).

Notice that the complexity of the calculation described above is the same as the complexity of the inputs $f$ and $g$. The particular solution $u_p$ is directly computed using the precomputed algebraic varieties, as described above and in Appendix B (fibers of the characteristic variety of $A$). Therefore, the cost of performing regression on the affine space (15) (inhomogeneous system) is the same as performing regression for the linear space (1) (homogeneous system) plus the cost of discretizing the inputs $f$ and $g$; recall here that both systems are underdetermined. In this sense, *B-EPGP does not suffer from the curse of dimensionality*.

We present an abridged version of this algorithm to use B-EPGP to generate solutions of (15).

1. Find a particular solution $u_p$ of (15) using another method, e.g. the one described above;
2. Use B-EPGP to generate $v$ such that $A(\partial)v = 0$ and $B(\partial)v = 0$;
3. Set $u = v + u_p$.

### M.1  2D WAVE EQUATION WITH INHOMOGENEOUS BOUNDARY CONDITIONS

In this section, we demonstrate how to conduct the reduction described above in the case of the 2D wave equation with inhomogeneous boundary condition. Let $\Omega = (0, 4)^2$. Here the inhomogeneous system reads

$$\begin{cases} u_{tt} - (u_{xx} + u_{yy}) = 0 & \text{in } (0, 8) \times \Omega \\ u(t, x, y) = f(t, x, y) & \text{on } \partial\Omega, \end{cases} \tag{16}$$

where we consider two cases. The first case $f(t, x, y) = f_1(t, x, y) = 0.1(x^2 + y^2 + 2t^2)$ and the second case $f(t, x, y) = f_2(t, x, y) = 0.5(x + y)$.

We observe that $u_p = f$ is a particular solution in both cases. We proceed with the homogeneous system

$$\begin{cases} v_{tt} - (v_{xx} + v_{yy}) = 0 & \text{in } (0, 8) \times \Omega \\ v = 0 & \text{on } \partial\Omega. \end{cases} \tag{17}$$

We would like to infer this solution using B-EPGP from some data. In this case, we will assume this data is sampled from some initial conditions,

$$\begin{cases} u(0, x, y) = 10\exp(-10((x - 2)^2 + (y - 2)^2)) & \text{in } \Omega \\ u_t(0, x, y) = 0 & \text{in } \Omega, \end{cases} \tag{18}$$

which then give initial conditions for $v$

$$\begin{cases} v(0, x, y) = 10\exp(-10((x - 2)^2 + (y - 2)^2)) - f(0, x, y) & \text{in } \Omega \\ v_t(0, x, y) = -f_t(0, x, y) & \text{in } \Omega. \end{cases} \tag{19}$$

We can now use B-EPGP to fit solutions of (17) to data sampled from (19) using the method described in Appendix G. If $v$ is our prediction, then $u = u_p + v$ is the prediction for our solution of (16) with data sampled from (18).

Snapshots of the solutions in both cases $f = f_1$ and $f = f_2$ are presented in Figure 17.

### M.2 2D HEAT EQUATION WITH INHOMOGENEOUS BOUNDARY CONDITIONS

In this section, we demonstrate how to conduct the reduction described above in the case of the 2D heat equation with inhomogeneous boundary condition. Let $\Omega = (0, \infty)^2$. Here the inhomogeneous system reads

$$\begin{cases} u_t - (u_{xx} + u_{yy}) = 0 & \text{in } (0, 0.2) \times \Omega \\ u(t, x, y) = f(t, x, y) & \text{on } \partial\Omega, \end{cases} \tag{20}$$

where we consider $f(t, x, y) = 10t^2 \sin(y) + 10t \sin(x) + 10t \cos(y) + 10t$.

In this circumstances, a particular solution can not be solved easily, we used the proposed spectral theory to find such a solution $u_p$. Then we proceed with the homogeneous system

$$\begin{cases} v_t - (v_{xx} + v_{yy}) = 0 & \text{in } (0, 0.2) \times \Omega \\ v = 0 & \text{on } \partial\Omega. \end{cases} \tag{21}$$

We would like to infer this solution using B-EPGP from some data. In this case, we will assume this data is sampled from some initial conditions,

$$u(0, x, y) = 10 \exp(-10((x-1)^2 + (y-1)^2)) \text{ in } \Omega \tag{22}$$

which then give initial conditions for $v$

$$v(0, x, y) = 10 \exp(-10((x-1)^2 + (y-1)^2)) - u_p(0, x, y) \text{ in } \Omega \tag{23}$$

We can now use B-EPGP to fit solutions of (21) to data sampled from (23). If $v$ is our prediction, then $u = u_p + v$ is the prediction for our solution of (20) with data sampled from (22).

The MSE of $u_p - f$ on $(0, 0.2) \times \partial\Omega$ is $3.45 \cdot 10{-}6$, whereas the MSE of $u - 10 \exp(-10((x-1)^2 + (y-1)^2))$ on $\Omega$ is $6.07 \cdot 10^{-6}$. This demonstrates that the error introduced by fitting inhomogeneous boundary conditions with our spectral method is on par with the error introduced by fitting initial conditions for problems with homogeneous boundary conditions. Therefore "inhomogeneous B-EPGP" does not incur unexpected costs over "homogeneous B-EPGP".

Snapshots of the solution in $[0, 3]^2$ are presented in Figure 18.

### M.3 LAPLACE EQUATION WITH A SINGULAR POINT

We also give an example with inhomogeneous boundary condition for Laplace's equation. In this case, we use EPGP to fit the boundary condition and also notice that the application of B-EPGP for the homogeneous system can only yield the zero solution.

By its very ansatz, EPGP produces global solutions, since each exponential-polynomial solution is a solution in full space (see also Appendix F). Here we demonstrate that our methods can approximate solutions on bounded domains $\Omega$ which cannot be extended to a global solution on $\mathbb{R}^n$. This somewhat surprising fact shows that B-EPGP approximates solutions *locally* very well.

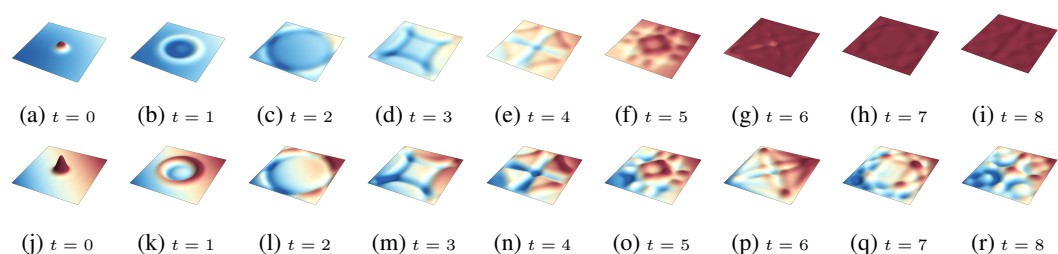

(a) $t = 0$   (b) $t = 1$   (c) $t = 2$   (d) $t = 3$   (e) $t = 4$   (f) $t = 5$   (g) $t = 6$   (h) $t = 7$   (i) $t = 8$

(j) $t = 0$   (k) $t = 1$   (l) $t = 2$   (m) $t = 3$   (n) $t = 4$   (o) $t = 5$   (p) $t = 6$   (q) $t = 7$   (r) $t = 8$

Figure 17: Solutions to the initial boundary value problem for the 2D wave equation with inhomogeneous boundary conditions. Notice that the height of the wave around the boundary is increasing over time. The boundary conditions in the first example are nonlinear. Animations can be found in the supplementary material as `2Dwave_nonhomo1.mp4` and `2Dwave_nonhomo2.mp4`.

Table 3: We demonstrate how good is our local approximation of a function that is not a global solution due to a singularity at $(0, 0)$. We use 10000 data points and $n = 10$ or $n = 50$ FAS elements. We emphasize that the error is really small, despite the distance between the domain and the singularity point being very small.

| Domain | Absolute L1 Error | | Relative L1 Error | |
|---|---|---|---|---|
| | $n = 10$ | $n = 50$ | $n = 10$ | $n = 50$ |
| $[1, 10]^2$ | 0.00030 | 3.6e-6 | 0.00036 | 3.59e-5 |
| $[1, 100]^2$ | 0.00081 | 7.68e-5 | 0.00079 | 6.93e-5 |
| $[0.01, 1]^2$ | 0.00875 | 0.00085 | 0.00642 | 0.00063 |

Let $\Omega \subset \mathbb{R}^2$ be such that $0 \notin \bar{\Omega}$. We consider the problem

$$\begin{cases} u_{xx} + u_{yy} = 0 & \text{in } \Omega \\ u = \log{(x^2 + y^2)} & \text{for } (x, y) \in \partial\Omega. \end{cases}$$

For this boundary condition, there is no $v$ satisfying $v_{xx} + v_{yy} = 0$ in $\mathbb{R}^2$ which satisfies the boundary condition. This is due to the uniqueness properties of *harmonic functions*: any such $v$ would have to equal $\log(x^2 + y^2)$ for $(x, y) \neq (0, 0)$, which is not defined in $(0, 0)$. This means that even though $\log(x^2 + y^2)$ satisfies the PDE in $\mathbb{R}^2 \setminus \{0\}$ as well as the boundary condition, our approximation must deteriorate as $\Omega$ is chosen closer to $(0, 0)$.

In Table 3 we report on our results with various domains at various different distances from the singularity $(0, 0)$ and the results are very good even when the distance is only .01. Figure 19 shows that the errors concentrate on the points that are nearer to the singularity of $\log(x^2 + y^2)$.

## N    EXPERIMENTAL DETAILS

In this appendix we collect all training details pertaining to the implementation of B-EPGP for our experiments, as well as for the algorithms we compare with (FNO, CNO, EPGP). All experiments are performed on an A100 Nvidia GPU with 80GB RAM.

### N.1    1D WAVE EXPERIMENT FROM SECTION 5.1

The neural operators were run on the reference code of the authors of (Raonic et al., 2024) and (Li et al., 2021) with their default settings. The most important are a learning rate 0.001, 50 epochs, stepsize 15, a channel multiplier of 16, and input and output spatial size of 256.

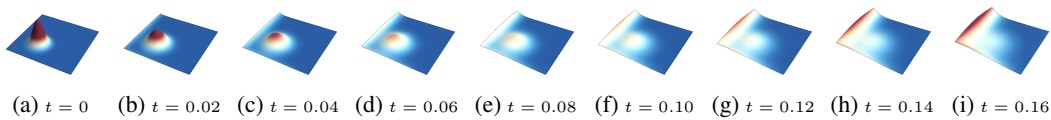

(a) $t = 0$    (b) $t = 0.02$    (c) $t = 0.04$    (d) $t = 0.06$    (e) $t = 0.08$    (f) $t = 0.10$    (g) $t = 0.12$    (h) $t = 0.14$    (i) $t = 0.16$

Figure 18: Solutions to the initial boundary value problem for the 2D heat equation with inhomogeneous boundary conditions. Notice that the height of the wave around the boundary is increasing over time. The boundary conditions in the first example are nonlinear. Animations can be found in the supplementary material as `2Dheat_nonhomo.mp4`.

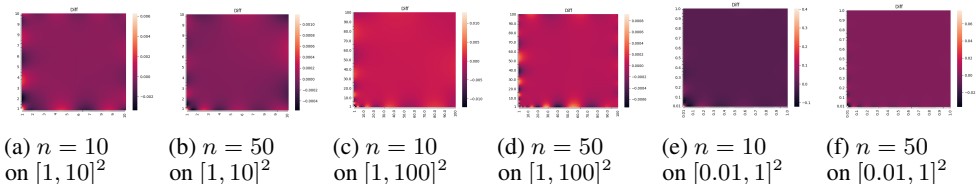

(a) $n = 10$ on $[1, 10]^2$    (b) $n = 50$ on $[1, 10]^2$    (c) $n = 10$ on $[1, 100]^2$    (d) $n = 50$ on $[1, 100]^2$    (e) $n = 10$ on $[0.01, 1]^2$    (f) $n = 50$ on $[0.01, 1]^2$

Figure 19: Error in our solution for Laplace's Equation in Section M.3. The error concentrates near the singularity at $(0, 0)$ and shows a wave-like behavior, similar to Gibbs phenomenon.

For B-EPGP and EPGP, we have 1000 FAS elements, whose prior is distributed as a $\mathcal{N}(0, I)$ distribution, first train 10000 epochs with learning rate of 0.1, then 10000 epochs with learning rate 0.01, then 1000 with 0.001. All training data and evaluation data are sampled with equal distance, while training data lies in the initial plane (and boundary if necessary), evaluation data lies in the whole space.

### N.2    3D WAVE EXPERIMENT FROM SECTION 5.2

We chose the number of FAS elements to be 500, whose prior is distributed as a $\mathcal{N}(0, I)$ distribution, first trained 10000 epochs with learning rate of 0.1, then 10000 epochs with learning rate 0.01, then 1000 with 0.001.

### N.3    2D WAVE WITH HYBRID BOUNDARY EXPERIMENT FROM SECTION 5.2

We chose the number of FAS elements to be 1000, whose prior is distributed as a $\mathcal{N}(0, I)$ distribution, first trained 10000 epochs with learning rate of 0.1, then 10000 epochs with learning rate 0.01, then 1000 with 0.001.

### N.4    FREE WAVE EQUATION FROM APPENDIX G

We chose the number of FAS elements to be 2000, whose prior is distributed as a $\mathcal{N}(0, I)$ distribution, first trained 10000 epochs with learning rate of 0.1, then 10000 epochs with learning rate 0.01, then 1000 with 0.001.

### N.5    2D WAVE WITH DIFFERENT DOMAINS FROM APPENDICES I AND K

We chose the number of FAS elements to be 1000, whose prior is distributed as a $\mathcal{N}(0, I)$ distribution, first trained 10000 epochs with learning rate of 0.1, then 10000 epochs with learning rate 0.01, then 1000 with 0.001.

### N.6    COMPARISON BETWEEN B-EPGP AND EPGP ON 2D WAVE FROM APPENDIX J

For B-EPGP, we chose the number of FAS elements to be 1000, whose prior is distributed as a $\mathcal{N}(0, I)$ distribution, first trained 10000 epochs with learning rate of 0.1, then 10000 epochs with learning rate 0.01, then 1000 with 0.001.

For EPGP, we chose the number of FAS elements to be 2000, whose prior is distributed as a $\mathcal{N}(0, I)$ distribution, first trained 10000 epochs with learning rate of 0.1, then 10000 epochs with learning rate 0.01, then 1000 with 0.001.

### N.7    2D HEAT EXPERIMENT FROM APPENDIX L

We chose the number of FAS elements to be 1000, whose prior is distributed as a $\mathcal{N}(0, I)$ distribution, first trained 10000 epochs with learning rate of 0.1, then 10000 epochs with learning rate 0.01, then 1000 with 0.001.

### N.8    INHOMOGENEOUS BOUNDARY EXPERIMENT FROM APPENDIX M.1

When training the homogeneous system, we chose the number of FAS elements to be 1000, whose prior is distributed as a $\mathcal{N}(0, I)$ distribution, first trained 10000 epochs with learning rate of 0.1, then 10000 epochs with learning rate 0.01, then 1000 with 0.001.

