# OpenReview forum: "Gaussian Process Priors for Boundary Value Problems of Linear Partial Differential Equations"
_ICLR.cc/2026/Conference — Submitted to ICLR 2026_

### Official Review · Reviewer_2LpB · 2025-10-20

**Soundness:** 3
**Presentation:** 3
**Contribution:** 3
**Rating:** 6
**Confidence:** 3

**Summary:**

The proposed methodology makes use of Ehrenpreis–Palamodov boundary-repecting basis functions to construct highly constrained Gaussian Process prior for PDE inference. The method is primarily used on the wave equation, showing its high accuracy. The method is compared to neural operators, 3D wave equation, and shown to be used with circular sectors.

**Strengths:**

- The paper is written with good precision.

- The proposed methodology is theoretically sound and mathematically grounded, answering one of the principle challenges in applying ML to problems in PDEs.

- Great care is taken in proposing a principled approach.

- Extensive exposition and testing is provided in the appendix.

**Weaknesses:**

- Much attention is given to describing how to constrain basis functions to respect boundary conditions. Could some amount of space be dedicated to showing how this is incorporated into the GP? i.e. how do we assign variances to each basis term?

- Arbitrary polygonal boundaries are discussed. These are of great importance in engineering applications. The construction of the basis seem to be carried out by hand. Could it be stated more clearly if there is an algorithm for computing these bases? Including pseudocode to this end would be helpful. If such an algorithm does not exist, how could one go about constructing an approximate automated method for this?

- Handling boundary conditions is the principle objective of this paper, however, only simple boundaries and geometries are considered. For practical relevance, it would be important to include an example of non-trivial geometry.

- How does this methods fair with the incorporation of observational data; it seem the method is only used for solving PDEs?

**Questions:**

- Can more clarity be given to (1) and explain the relevance of half spaces here and how this relates to the PDEs of interest? Perhaps some foreshadowing could help readers navigate this section.

- As the PDE is assumed to be known as well as the boundary conditions, can this problem be rephrased as an inverse problem for recovering initial conditions? Is so, stating this explicitly would be helpful.

---

> ### Author Response · Authors · 2025-11-21
> **Rebuttal to Reviewer 4**
>
> We thank Reviewer 4 for constructive questions regarding our draft, which lead to some improvements.
>
> > Much attention is given to describing how to constrain basis functions to respect boundary conditions. Could some amount of space be dedicated to showing how this is incorporated into the GP? i.e. how do we assign variances to each basis term?
>
> Once basis functions are chosen, our approach uses them just as Bayesian linear regression works. We describe this in Appendix F (formerly E).
>
> > Arbitrary polygonal boundaries are discussed. These are of great importance in engineering applications. The construction of the basis seem to be carried out by hand. Could it be stated more clearly if there is an algorithm for computing these bases? Including pseudocode to this end would be helpful. If such an algorithm does not exist, how could one go about constructing an approximate automated method for this?
>
> We have added a new appendix (C) to make our approach more explicit.
>
> > Handling boundary conditions is the principle objective of this paper, however, only simple boundaries and geometries are considered. For practical relevance, it would be important to include an example of non-trivial geometry.
>
> We understand the concern about practical geometries. While many real-world domains are indeed complex, our current examples already go beyond the simplest cases: they include rectangles, slabs, wedges, circular sectors, etc., which are standard boundary configurations in engineering. We also added further non-trivial geometries in Appendices M.1 and M.2.
>
> > How does this methods fair with the incorporation of observational data; it seem the method is only used for solving PDEs?
>
> Our method is a regression methods, which means you can feed with any observation data (not necessary at t=0) and find a solutions that fits the PDE as well as boundary conditions. Operationally, we treat B-EPGP as a standard GP regression model: given observations $\{(x_i,y_i)\}$ at arbitrary space–time locations, we condition the prior on these data to obtain a posterior over solutions that satisfy the PDE and boundary conditions everywhere. All experiments except Fig. 1 follow this pattern.”
>
> > As the PDE is assumed to be known as well as the boundary conditions, can this problem be rephrased as an inverse problem for recovering initial conditions? Is so, stating this explicitly would be helpful.
>
> Our approach also applies to inverse problems; see Section 4 (‘EPGP has recently been generalized to inverse problems (Li et al., 2025)’). We note, however, that this has so far been used for parameter inference, not for recovering initial conditions.
>
> Our method is a regression method, which means you can feed it with observation data at arbitrary space–time locations (not necessarily at $t=0$) and obtain a solution that fits both the PDE and the boundary conditions. This is what we do in all experiments in the paper and appendices, with the only exception of (the more illustrative) Figure 1.

---

> > ### Comment · Reviewer_2LpB · 2025-11-24
> > **Response to rebuttal**
> >
> > Thank you for answering my questions and comments. I will keep my score as it is.

---

### Official Review · Reviewer_r3kn · 2025-10-30

**Soundness:** 2
**Presentation:** 1
**Contribution:** 2
**Rating:** 2
**Confidence:** 4

**Summary:**

**(Nb: when I submitted this review, curly braces did not render. If they are not visible in the final review, please read the text as if there were braces around the sets. If they are visible, ignore this message.)**


The submission is the next iteration in the work of Gaussian process priors based on Ehrenpreis-Palamodov theory. More concretely, it extends the work by Härkönen et al. (2023), who develop Gaussian process priors that satisfy linear PDEs exactly (but no boundary conditions) to priors that satisfy linear PDEs and linear boundary conditions exactly. The approach goes (roughly) as follows:
- Solutions of linear ODEs with constant coefficients can be expressed in terms of basis functions: e.g., for $y'' - 3y' + 2y = 0$, we have $y = w_1 e^t + w_2 e^{2t}$ for some coefficients $w_1$ and $w_2$. Letting $w_1$ and $w_2$ be Gaussian variables, this $y$ becomes a parametric Gaussian process (GP) constrained to the linear ODE.
- For partial differential equations (PDEs), a similar setup is possible, but it is more involved in that the basis functions are an uncountably infinite set that needs truncation. For example, for the heat equation $u_t = u_{xx}$ in 1d, any solution can be approximated arbitrarily well by a (complex) linear combination of finitely many elements of $\{e^{z^2 t + zx} \mid ~ {z \in \mathbb{C}}\}$, which is due to the Ehrenpreis--Paladomov theorem. Choosing a finite number of such $z \in \mathbb{C}$, and linearly combining them with Gaussian weights, we get a parametric GP constrained by the heat equation. The hyperparameters of this parametric GP can be optimised with the usual Gaussian process machinery. This has been done by Härkönen et al. (2023).
- The submission now extends this to the case of linear PDEs with linear boundary conditions. This is based on the idea that for a finite selection of $z \in \mathbb{C}$, which leads to a finite number of functions, the weights can be constrained to automatically satisfy the boundary conditions. For the heat equation in 1d and boundary condition $x=0$, we get the collection $\{e^{z^2t + zx} - e^{z^2t - zx} \mid z \in \mathbb{C} \}$. The submission also discusses wave equations, Neumann conditions, and higher-dimensional settings, and it is shown that the restricted candidate functions (eg $\{e^{z^2t + zx} - e^{z^2t - zx} \mid z \in \mathbb{C} \}$) are dense in the space of solutions of each respective PDE of interest, similar to the original Ehrenpreis--Paladomov theorem.

Numerical results show that this closed-form enforcement in the proposed method beats not enforcing the boundary conditions in closed form, which is necessary if one relies on previous work.


**Summary of my recommendation:**
I find this work promising, but the paper seems incomplete, by which I mean that the method and the experiments need a more careful explanation, and that the experiments miss some important questions. Concrete examples follow under "Weaknesses" below. In general, I believe that the algorithm has potential, but I find the current version not sufficiently accessible, not even to the subset of the machine-learning community that focuses on the intersection of differential equations and GPs (like myself). Therefore, I recommend rejection.

**Strengths:**

In principle, I really like the line of work that derives parametric GPs so that they automatically satisfy PDEs, including the submission, but also prior work by Härkönen et al., Lange-Hegermann, and others. What I like especially about the submission and the closely related Härkönen et al. is that it applies to underspecified PDE problems. Enforcing essentially unique functions would not give much space for machine-learning algorithms to, loosely speaking, "do their thing".
As such, I believe the setup in this submission fits well into the literature on physics-informed machine learning.
And even though I have strong enough concerns about the presentation to recommend rejecting this work, I expect a future version of this manuscript to be a valuable contribution to the machine learning literature.

**Weaknesses:**

Even though I think that the general idea of the contribution has potential, I think that the submission has considerable gaps in presentation and numerical evaluation:

**A lack of mathematical precision inhibits readability:**
It is clear from reading the manuscript that the authors are skilled mathematicians; however, the presentation lacks the technical precision that I would expect from this kind of work:

- When writing "linear span" (eg in Theorems 2.1 or 3.6), do I assume correctly that we mean linear combinations with complex coefficients? Distinguishing real and complex coefficients is important because complex linear combinations would stand in contrast to the real-valued Gaussian coefficients used to define a parametric Gaussian process (line 106, line 168). In Line 144, it would be good to specify whether $w_j$ are some fixed real/complex coefficients or whether they are Gaussian random variables, and it would also be good to explicitly state that the number and location of the $z_j$ are chosen by users. After reading the full manuscript, I have answered these questions myself, but I would prefer if they were unambiguous at the first occurrence. In general, since Ehrenpreis--Paladomov theory is not trivial, I would recommend carefully defining each variable at each occurrence (eg the imaginary unit in Line 223 comes a bit out of nowhere), even if this makes the notation in this paper somewhat heavier.

- I think it would help accessibility if the word "basis" were used a bit more carefully: For example, why would it be clear (at this point of the manuscript) that Equation 3 is indeed a basis, which line 164 states, but only Theorem 3.6 shows?
Similarly,  the manuscript regularly calls a set of functions "basis" before the theorem that shows this set of functions spans a space of interest (eg in Lines 223 right before Theorem 3.9; also, linear independence is not discussed but I think it should be). I recommend being more precise with which features are known about a set of functions at the time of introducing it, eg in line 164, line 215, line 233, but also on other occasions.

- The submission contains three types of sets of functions that serve the purpose of function approximation. There are finite expansions (eg $f$ in line 144), uncountably infinite sets of functions (Theorem 3.6) and countably infinite sets of functions (Theorems 3.9 and 3.11). It is not stated explicitly that the finite expansion in Line 144 serves only as intuition for what follows after, determining what eventually leads to Equation 3. Furthermore, the step from uncountably infinite to countably infinite sets of functions in line 224 needs to be explained more thoroughly, both in the main paper and in Appendix J.1 (eg that this set is a basis is only determined later). It would be nice if a future version of this manuscript had a longer explanation at these links.

I understand that the level of technical rigour in machine learning is a bit lower than in mathematics publications. However, making the technical derivations in Section 3 more precise would make the paper easier to understand.
To realise these suggestions, I would recommend showing fewer examples in the main paper but explaining each of them in more detail. For example, I found Appendix C to be required reading for understanding Section 3, and if Example 3.8 and Theorem 3.9 are explained thoroughly, Example 3.10 and Theorem 3.11 could almost be left to the reader.



**Important numerical considerations are omitted:**
In my opinion, the paper does a good job of deriving those sets of functions that span the spaces of PDE solutions. However, computational considerations are not discussed adequately: the only mention of constructing a Gaussian process model is in Line 105, and this reference only mentions optimising $z_j$ and the hyperparameters of the prior over each $w_j$ with stochastic gradient descent. The submission does not explain which loss is used. The choice of loss matters because while hyperparameters for $p(w_j)$ may be straightforward to optimise with marginal likelihood calibration (typical for GPs), optimising $z_j$ is closer to inducing points respectively GP latent variable models, for which the state of the art uses more sophisticated techniques (typically, based on variational inference). Independent of what is actually used in the experiments, not discussing these considerations at all is a major shortcoming of the submission.
The accuracy of the regression tasks (and difficulty of the optimisation problem) also depends on the number and location of basis functions, and the effects of this approximation need to be studied, at least empirically. Regarding baselines, the present work only compares to Härkönen et al. (and neural operators), but other GP-PDE combinations are not benchmarked. For example, I would have expected physics-informed GPs to be a baseline, possibly in combination with inducing points; something like:

> Hamelijnck, Oliver, Arno Solin, and Theodoros Damoulas. "Physics-informed variational state-space Gaussian processes." Advances in Neural Information Processing Systems 37 (2024): 98505-98536.

As is, the submission does not embed as well into the literature on GPs and PDEs as is necessary for me to recommend publication at ICLR.



**Related work:**
Finally, I would like to discuss the choice of related work mentioned in the submission. I consider the following weakness significantly less important than the ones above, but I would like to mention it regardless.
I appreciate the thorough related work section, but I think that in the current version, it optimises a bit too much for breadth and too little for depth. For example, other than also belonging to the class of PDE-ML algorithms, I am a bit surprised that neural operators and PINNs feature prominently in the related work. I understand the desire to make the work accessible to a broad audience at ICLR, but currently, this is to the detriment of discussing closely related work adequately. Instead of neural operators and PINNs, I would have appreciated more depth on the explanation of how the present work differs from Lange-Hegermann (2018 & 2021), Jidling et al. (2017), and the rest of the citations in Lines 042. I also think some additional references might deserve closer discussion:
Latent force models (Alvarez et al.), especially in their SDE variation (Hartikainen et al.), also construct ODE/PDE-informed priors for GP regression. They're doing something slightly different from the proposed work, but I think they're more closely related than many of the works cited in Section 4. See also Bosch et al., who use these linear-ODE priors for probabilistic solvers for nonlinear ODEs:

> Alvarez, Mauricio, David Luengo, and Neil D. Lawrence. "Latent force models." Artificial intelligence and statistics. PMLR, 2009.

> Hartikainen, Jouni, and Simo Särkkä. "Sequential Inference for Latent Force Models." Proceedings of The 27th Conference on Uncertainty in Artificial Intelligence (UAI 2011), Barcelona, Spain, July 14-17, 2011. 2011.

> Bosch, Nathanael, Philipp Hennig, and Filip Tronarp. "Probabilistic exponential integrators." Advances in Neural Information Processing Systems 36 (2023): 40450-40467.

**Questions:**

The following are minor questions and comments that do not really affect my score, but might be helpful to incorporate into a revised version of this manuscript.

- Line 40: What is meant by "bilinear covariance structure" in this context?
- Line 104: I appreciate that the main paper explains the concepts using a simplified setting of Ehrenpreis--Palamodov theory. It would be nice if the manuscript could discuss to what extent the simplifying conditions of Theorem 2.1 hold. For example, I find it non-obvious whether the characteristic equations of an ODE have a multiplicity (eg the example given in Line 090 does), but for all PDEs studied in this submission, multiplicities seem to be no issue.
- Line 162: I understand that Gröbner bases were important for prior work in this space, but in the present submission, all bases were possible to derive with pen and paper (as is stated in Line 815). Since terminology like "Gröbner basis" or "syzygy module" is likely not common vocabulary at ICLR, why are lines 163 and Appendix B important for this submission?
- The submission regularly states that traditional GP methods and traditional PDE solvers suffer from the curse of dimensionality (eg Line 127, Line 409) and that the proposed algorithm does not. I am wondering whether more nuance is required here, because while the proposed method does not need to place a grid to enforce boundary conditions, the number of terms in the basis functions grows with dimension. For example, compare Example 3.7 (2d) to Example 3.8 (1d). Also, the derivation of basis functions becomes more laborious in higher dimensions. Line 953 states:

    > The calculations extend easily to arbitrary dimensions, but the formulas are cumbersome

    which makes it sound like extension to arbitrary dimensions is not as easy in practice. I recommend being more careful with claims about avoiding the curse of dimensionality (I find the contribution of exact enforcement strong enough on its own).

---

> ### Author Response · Authors · 2025-11-21
> **Rebuttal to Reviewer 3**
>
> We thank Reviewer 3 for a *very objective* and constructive review that clearly aims at *improving* the paper. We appreciate your opening remark that you find the work promising, as well as your recognition that the related-work section is thorough and that the manuscript demonstrates a strong mathematical background, even if the exposition initially lacked the precision and accessibility you were hoping for. We are also grateful for your positive comments on the simplified Ehrenpreis–Palamodov presentation in the main text. Your detailed suggestions on terminology, numerical methodology, and positioning within the GP–PDE literature have served as a roadmap for our revision. We believe that your concerns can be addressed by focused and clear changes to the main paper and have revised the paper in this direction.
>
> ## Regarding the terms "linear span", "basis", their cardinality, and the question about real and complex coefficients
>
> We attempted to implement all of the reviewer's suggestions regarding mathematical notation and explanations, please see the updated version.
>  - We replaced the term "basis" with "FAS'' (Finitely-Approximating Set, Section 2), a (finite or countable) set whose complex linear span is dense in the solution space endowed with the standard Fréchet topology. Linear independence is irrelevant for us, which makes FAS exactly the object we need. In implementation we always work with finite truncations of such sets.
>  - We clarified that for the algebraic calculations we are using *complex* values everywhere (except $x in \Omega\subset R^n$), whereas we use *real* GPs in the statistics part. We take real/imaginary parts to pass from complex algebra to real statistics. We made this clear in a new Section 3.4 and computed a very explicit and detailed example in Appendix I.3.
>  - Concerning finite/countable/uncountable FAS: We think it is very clear that the implementation works only with finite sums, approximating infinite dimensional solution spaces. So all our implementations naturally use only finite sums. We expanded on the cases where only countably many parameters $z_j$ are necessary for B-EPGP around Theorem 3.9. The argument is: we observe countable FAS candidate and we prove candidate is FAS by Fourier Analysis.
>  - We inserted important calculations from Appendices into Section 3.1, which is the core of our novelty.
>  - We made many minor modifications according to the Referee's careful and thoughtful suggestions.
>
> ## Regarding the loss function, ablations w.r.t. number of terms, and comparison to other physics informed GP papers
>
> The loss function we used was described in Appendix F (formerly E) and is now referenced more prominently in the main paper.
>
> We did conduct experiments in our original submission, which describes how the number of basis functions affects regression accuracy. There results are shown in Appendix J (formerly I) in Figure 13. We have now referenced this appendix more prominently in the main paper.
>
> Regarding comparing to physics-informed GPs, we do compare to two GP based methods both BCGP (JMLR 2024) and WIGPR (Journal of Computational Physics, 2023), which are both recent GP-PDE methods in relevant problems. We did make the comparison the GP methods  more clear in the corresponding subsection title. We did not compare to Hamelijnck et al. (2024), because BCGP and WIGPR are more focused on the PDEs we wanted to solve (linear, boundary conditions), whereas Hamelijnck et al. (2024) focus more on the evolution aspect of partial differential equations. Both paper are from the same time and similarly prominently published.
>
> We did not compare to inducing point methods, as our approach is computationally efficient enough to do without such approximations.
>
> ## Regarding depth and breadth of our description of the state of the art
>
> After about a page and 90 references in total, we put a stop to discussing every paper and every detail. We acknowledge that we might have put our focus on discussing the wider literature. This is addressed in the current version of the paper, which now has 99 references and puts a stronger focus on GP methods for PDEs. We now clearly distinguish B-EPGP from collocation-based GP-PDE methods by emphasizing that our priors encode PDE + boundary constraints analytically rather than through pointwise conditioning.

---

> > ### Author Response · Authors · 2025-11-21
> > **Rebuttal to Reviewer 3 - Minor Questions**
> >
> > ## Line 40: What is meant by "bilinear covariance structure" in this context?
> >
> > "Bilinear" was meant in the trivial sense that the covariance is bilinear in its arguments, which is standard for GP kernels. Since this wording is unnecessary and potentially confusing, we have removed “bilinear” and now simply refer to the “covariance structure” of the GP in the revised manuscript.
> >
> > ## Regarding multiplicities
> >
> > For ODEs, multiplicities correspond to multiple roots of the characteristic polynomial (as in the toy example around line 90, which has a multiple root at $z=1$. For PDEs, true multiplicities in the sense of primary decomposition of the characteristic variety are much rarer and more subtle to describe. To keep the main text readable we focus there on the simple irreducible, multiplicity-free case of Theorem 2.1, and move the general case (with multiple components and multiplicities, including the polynomial “correction factors”) to Appendix E.
> >
> > ## Regarding the importance of Gröbner bases
> >
> > You are right that all examples shown in the main text can be derived by hand. We still mention Gröbner bases and syzygy modules because they provide a general symbolic algorithm for constructing and certifying boundary-respecting bases beyond these toy cases (e.g. Maxwell, elasticity), where manual derivations quickly become tedious. The main text only “advertises” this capability (with minimal notation), while Appendix B contains the full algebraic treatment and concrete Singular code. This also connects our work with the broader symbolic-computation literature underlying earlier EPGP-type methods.
> >
> > ## Regarding the curse of dimensionality
> >
> > Here we mean simply that we pay no cost from fitting at the boundary: If we fit $u(x,0)=0$ with x in $R^{n-1}$, the cost we incur is 0, independently of $n$. The formulas for higher dimensions are cumbersome by hand, but not cumbersome for a computer: in Appendix B we explain that the task of calculating B-EPGP FAS is solvable using existing Computer Algebra tools. Of course, there is a price to pay with increasing dimension: take the example in Remark 3.1. If $x\in R^D$ is taken from a higher dimensional space, our computations will scale with $D$ when fitting the initial condition $f(x)$; but the cost will not scale with taking the cost of fitting a boundary condition $u=0$ at say $x_1=0$, a $D$-dimensional condition also; much like we don't pay to satisfy the equation in $D+1$ dimensions. As for increasing the number of FAS elements, there is a very recent comparison between B-EPGP and classical numerical methods under comparable number of features (degrees-of-freedom), which shows that for 2D wave equation, B-EPGP performs 2-3 orders of magnitude better. A preprint is available at arXiv:2511.04518. In response to your concern, we have weakened our statements about the ‘curse of dimensionality’ and now state more precisely that B-EPGP avoids boundary grids in any dimension, while retaining the usual dimension-dependent costs for representing and conditioning on interior data.

---

> > > ### Comment · Reviewer_r3kn · 2025-11-26
> > >
> > > Thank you for the detailed reply.
> > >
> > > - Linear span and basis: thank you, I think these changes make the presentation a lot better.
> > > - Loss functions and ablations: Thank you for mentioning these parts more prominently. I think they might have gotten lost in the long supplement; apologies for stating they do not exist.
> > > - Depth and breadth: This might be a misunderstanding. My original review did not state that the submission lacks references (or volume), and I hope that it did not read as such -- quite the contrary: I was surprised to see neural operators and PINNs to feature that prominently, and I would have been okay with a related work section that does not feature neural operators at all, for example. What the review actually meant was that the choice of which related work is discussed in depth and which isn't did not match my expectations: I expected in-depth discussions of how the submission compares to GP methods, especially those that construct GPs that satisfy PDEs exactly, and at most shallow references to everything else. The submission chose a different emphasis, which is why I mentioned related work under weaknesses. This is not something to correct fully during the rebuttal period, so I do not expect significant changes here, but I think it remains a weakness (though I appreciate the GP paragraphs in the revised documents). However, not all weaknesses must always be resolved fully before considering a paper publishable.
> > > - Curse of dimensionality: thank you for these revisions. I think it is indeed more accurate to write that the proposed method avoids boundary grids because "avoids the curse of dimensionality" is a strong claim. Where in the paper can one find these revisions? I still see many statements like "avoids the curse of dimensionality", e.g. in the beginning of Section 3 or line 2018.

---

> > > > ### Author Response · Authors · 2025-11-27
> > > >
> > > > We thank you for your time and your timely and friendly answer, which allows further clarifications!
> > > >
> > > > Regarding the literature, we appreciate this clarification. In the camera-ready version, we will revise the related-work section to focus more strongly on Gaussian-process–based methods, in particular those that enforce the PDE exactly, and we will streamline the discussion of PINNs and neural operators to brief contextual references only. This reallocation of space will allow us to give a much more detailed and nuanced comparison between B-EPGP and existing GP approaches.
> > > >
> > > > Regarding the curse of dimensionality, let us clarify what B-EPGP can do: (i) for 0 boundary conditions, there is **no additional cost** of fitting the boundary condition in polygonal domains, independent of dimension. (ii) for nonzero boundary conditions $f$, we explain in Appendix M that the cost of incorporating boundary conditions **equals the cost of discretizing $f$** (same order of magnitude, demonstrated in a new experiment). To discretize $f$, we use a meshless spectral method, which enables us to obtain a particular solution by direct calculation. In light of this, we will edit the camera-ready version around each current mention (l.128, l.2018) to say that we deal with the curse of dimensionality in this asymptotically optimal way.

---

> ### Comment · Reviewer_r3kn · 2025-11-28
>
> Thank you for elaborating. Assuming all promised changes will be included in the final manuscript, I will update my score.
>
> (Edit: I don't seem to be able to edit my review at the moment. I will try again later.)

---

### Official Review · Reviewer_yyWM · 2025-10-30

**Soundness:** 4
**Presentation:** 4
**Contribution:** 3
**Rating:** 8
**Confidence:** 4

**Summary:**

This paper proposes a Gaussian process method for solving boundary value problems of linear partial differential equations with constant coefficients. The key contribution is a framework that encodes boundary conditions directly into basis elements, allowing the construction of GP priors whose domain automatically aligns with the PDE solution space. By imposing Gaussian priors on parameters of the solution's functional form (where the functional form is known but parameters are unknown), the authors obtain a Gaussian process that exploits the specific PDE structure. The paper departs from standard collocation approaches and includes proofs of correctness and convergence. The authors also propose a hybrid approach that combines their method with data-driven techniques for handling curved boundaries.

**Strengths:**

## Strengths

1. **Novel GP construction**: The approach cleverly constructs kernels (from basis expansion) so that the GP domain naturally overlaps with the PDE solution space, which is an elegant solution to the problem.

2. **Strong theoretical contributions**: The paper provides rigorous derivations, theorems, and proofs (including proofs of correctness and convergence), which is commendable as theory often lags behind in this field. [however I didn't check the proof details]

3. **Exploits problem structure**: Unlike PINNs and collocation-based GP methods, this approach makes significant and intelligent use of the linear PDE structure, which should lead to better performance for the target class of problems.

4. **Clear exposition**: The examples (particularly at lines 90, 171-178) effectively communicate the intuition behind the method and help readers understand the approach.

5. **Solid work within scope**: For the specific area of linear PDEs with constant coefficients and limited data, this is useful and well-executed research.

**Weaknesses:**

## Weaknesses

1. **Requires bespoke treatment**: The method requires tailored, hand-crafted basis functions for each different PDE, with most calculations done manually. This is a stark departure from methods like PINNs that can be applied more automatically. While this leads to better performance, it significantly limits ease of use and scalability.

2. **Limited scope**: The method explicitly targets linear PDEs with constant coefficients, which is a restrictive problem class.

3. **Hybrid approach undermines main contribution**: Section 3.3's hybrid approach (line 255) doesn't integrate well with the elegant exact solution framework. If data is needed for curved boundary pieces, the claim of exactness is compromised, which diminishes the value proposition.

## Minor points
4. **Computational cost unclear**: For the hybrid approach (Section 5.3, line 420), it's unclear whether the method still suffers from the curse of dimensionality mentioned in Remark 3.1. The computational cost analysis needs clarification.

5. **Missing references**: Several relevant Gaussian process methods for PDEs should be cited (some suggestions have been provided by the reviewer). It will be better to position the work relative to existing Gaussian process techniques for PDEs (e.g., work by Andrew Stuart, Houman Owhadi, Mark Girolami, Philipp Hennig, Jon Cockayne). The novelty is unclear without reading deeper into the paper.

**Questions:**

I have listed structured questions (with help of LLM) in the above weakness part. I am going to say here my honest thoughts when reading the paper as it presents, and hopefully this can help you understand how a new reader perceives your paper. These raw feelings are genuine and I hope they provide a more human-to-human communication and contexts for the structured question above.

# Review of "Gaussian Process Priors for Boundary Value Problems of Linear Partial Differential Equations"


## Initial Impressions (Abstract)

I'm reviewing this paper on Gaussian process priors for boundary value problems of linear partial differential equations. From the name "EP”, the approach seems to be specific to linear PDEs.

The abstract feels very short and doesn't touch upon existing Gaussian process techniques, of which I'm aware of quite a few. After reading the abstract, I'm curious to see how this differs from the Gaussian process work by Andrew Stuart and the Houman Owhadi group at Caltech, and also from the probabilistic numerics group in the UK (Mark Girolami, Philipp Henning, and Jon Cockayne). The novelty isn't very clear from the abstract alone.

## Early Sections (Lines 31-48)

After reading line 37, I see that the authors are moving away from the collocation approach, which is interesting.

At the end of line 48, I think citations are needed. Most Gaussian process methods for solving PDEs still rely on certain characterizations of the PDEs that are not exact—they are exact at the collocation points where the information operator is conditioned, but the notion of "physics constraint" needs clarification. Please provide references to specify exactly which approaches you're discussing.

My understanding is that for the linear case, collocation GP also works really well, and the problem is basically solved. So I don't see what's new from line 31 yet.

Line 76 Point number three—the proof of correctness and convergence—is intriguing. In most of the work I've seen, the theory always lags a little behind. I'm curious what new contribution is introduced here.

Line 78 I also wonder why this is framed as a regression model. Neural operators learn many instances, so I need to see whether you're doing single-instance inference or many-instance inference.

## Problem Formulation (Line 90)

I really like line 90. The example helped me understand the problem being solved here. This is interesting: the functional form of the solution is known, but some parameters of this functional form are unknown. By imposing a Gaussian prior on these parameters, you effectively get a Gaussian process. This feels like a very smart construction of the kernel so that the solution space automatically aligns with the domain of the Gaussian process. I wouldn't be surprised if this performs better than PINNs because it exploits the particular PDE structure. Essentially, you've structured a specific Gaussian process that is engineered to overlap with the PDE solution space.

## Key Contribution (Lines 111-130)

Line 111 is the real contribution and the key idea. Now it's very clear. Line 130 further explains the problem from line 111: the boundary conditions are encoded into the basis elements.

After reading Equation 3, I understand the intuition. The boundary conditions can be encoded as basis functions, but this requires tailored or bespoke treatment for each PDE. You need to be clever in finding these bases.

The examples from lines 171 to 178 are very helpful in understanding the intuition. I wish there were a more automatic way to do this. I understand that right now, most of the calculation is done by hand. But this is a stark difference from PINNs, right? PINNs can be used very easily—they basically remove a lot of the necessity for smart calculations. There are pros and cons. I wouldn't be surprised if, after all this basis construction work, the performance is better. That's a no-brainer.

## Section 3.2 and Beyond

I'm being quick in Section 3.2 because I think I understand the gist of what's going on. I'm unable to check all the exact detailed mathematics, but at this point, I have enough intuition. You also claim there are many proofs in the appendix that I wasn't able to check.

In Section 3.3, I didn't understand in what sense the method is "hybrid." Is it hybrid because you have both flat and curved pieces of boundary? If you're using data for the curved pieces, then you've somewhat lost your claim of things being exact. I think Sections 3.3 actually diminish the value proposition—you can plug in collocation points, but it doesn't integrate quite as well with the elegant solution you're proposing. This comment refers specifically to line 255.

## Literature Review

For the literature search, since you mentioned PINNs and Gaussian processes, I think neural operators aren't that related to this work, but Gaussian process methods used in the PINN style should be referenced. I see you already have a few familiar names here: Philipp Hennig, Mark Girolami's group, Philipp Wenk, Andrew Stuart, Houman Owhadi, and some others. I'll suggest a few representative papers that I think you should cite as well.

However, your approach is significantly different from these. Those are Gaussian process counterparts of PINNs. Yours is very different—it makes significant use of the linear problem structure.

## Experiments

For the experiments section, I wouldn't be surprised by the results, given such heavy use of the problem structure and the mathematics to make it work. You do have experiments on the hybrid approach, but the hybrid approach doesn't quite align with your main storyline.

In Section 5.3, line 420: What's the computational cost of your hybrid approach? I would imagine your hybrid method still suffers from the curse of dimensionality mentioned in your Remark 3.1, unless I'm missing something.

## Overall Assessment

This is good work—very solid. I appreciate that you have all the derivations, examples, and theorems. The scope is limited, as it explicitly targets linear partial differential equations with constant coefficients. For that specific area where you have some data, I think this is useful.


## References

Cockayne, J., Oates, C., Sullivan, T. and Girolami, M., 2017, June. Probabilistic numerical methods for PDE-constrained Bayesian inverse problems. In AIP Conference Proceedings (Vol. 1853, No. 1, p. 060001).

Chen, Y., Hosseini, B., Owhadi, H. and Stuart, A.M., 2021. Solving and learning nonlinear PDEs with Gaussian processes. Journal of Computational Physics, 447, p.110668.

Wenk, P., Gotovos, A., Bauer, S., Gorbach, N.S., Krause, A. and Buhmann, J.M., 2019, April. Fast Gaussian process based gradient matching for parameter identification in systems of nonlinear ODEs. In The 22nd International Conference on Artificial Intelligence and Statistics (pp. 1351-1360). PMLR.

Li, Z., Yang, S. and Wu, C.J., 2024. Parameter inference based on Gaussian processes informed by nonlinear partial differential equations. SIAM/ASA Journal on Uncertainty Quantification, 12(3), pp.964-1004.

Long, D., Wang, Z., Krishnapriyan, A., Kirby, R., Zhe, S. and Mahoney, M., 2022, June. AutoIP: A united framework to integrate physics into Gaussian processes. In International Conference on Machine Learning (pp. 14210-14222). PMLR.

---

> ### Author Response · Authors · 2025-11-21
> **Rebuttal to Reviewer 2**
>
> We thank Reviewer 2 for the thoughtful and *very constructive feedback*, especially on how the writing and overall perception of the paper can be improved. We are particularly grateful for your positive remarks that the example around line 90 helped clarify the problem setup, and that line 111 (together with the explanation around line 130) made the key contribution—constructing a GP whose support aligns with the PDE solution space—feel clear and well-motivated. Your comments on the narrative structure have helped us a lot in streamlining about our paper, and we appreciate you taking the time to share this detailed, personal perspective.
>
> > Requires bespoke treatment: The method requires tailored, hand-crafted basis functions for each different PDE, with most calculations done manually. This is a stark departure from methods like PINNs that can be applied more automatically. While this leads to better performance, it significantly limits ease of use and scalability.
>
> We agree that our method trades some up-front effort in constructing a basis for large gains in accuracy and speed. Importantly, this basis construction is not purely manual: for the general constant coefficient case we provide an algorithm (fibers + syzygies) that automatically produces a boundary-respecting basis from the PDE and boundary operators, and for many common systems (heat, wave, elasticity, Maxwell) the resulting formulas are simple enough to be written down by hand. In practice this symbolic step is a one-time, PDE-specific, and fully automatic precomputation that is independent of the dataset, much like deriving a weak form in FEM; once the basis is available, training and inference are fully automatic.
>
> > Limited scope: The method explicitly targets linear PDEs with constant coefficients, which is a restrictive problem class.
>
> We agree that our current theory targets linear PDEs with constant coefficients. This class, however, already includes many central models in physics and engineering (wave, heat, Poisson, Maxwell, linear elasticity), as well as piecewise constant media where interfaces are modeled via additional boundary conditions. We now emphasize this more clearly in Section 4 and present our method explicitly as a high-accuracy, probabilistic tool for this important regime. Extensions to genuinely nonlinear problems (e.g., via wave-packet bases) are future work rather than as part of our present scope.
>
> > Hybrid approach undermines main contribution: Section 3.3's hybrid approach (line 255) doesn't integrate well with the elegant exact solution framework. If data is needed for curved boundary pieces, the claim of exactness is compromised, which diminishes the value proposition.
>
> First, in most scientific examples, domains are polygonal, often just rectangular.
> Second, there seems to be no way to extend our algorithm for halfspaces to general surfaces. This is so because the EPGP exponential bases are constant on hyperplanes and thus no *finite* linear combination can sastisfy linear conditions on a curved boundary (try solving $u=0$ on a circle).
> Therefore, proposed alternatives must resort to approximation. Practical solutions are possible:
> 1. Approximate the curved boundary with a polygonal one
> 2. The hybrid approach we propose in the paper, which delivers a stark improvement over sampling over the entire boundary (Fig. 3 and 4)
>
> > Computational cost unclear: For the hybrid approach (Section 5.3, line 420), it's unclear whether the method still suffers from the curse of dimensionality mentioned in Remark 3.1. The computational cost analysis needs clarification.
>
> We ran two experiments (EPGP vs B-EPGP) with the same number of parameters (basis functions). The gap in accuracy was similar to the one in Figure 3 (left). The running times were similar to those in Table 2: B-EPGP is about one order of magnitude faster than EPGP due to the reduced number of data points. We believe that the curse of dimensionality incurred by EPGP is greatly mitigated by using B-EPGP, while obtaining far sharper results. In response to your concern, we have weakened our statements about the ‘curse of dimensionality’ and now state more precisely that B-EPGP avoids boundary grids in any dimension, while retaining the usual dimension-dependent costs for representing and conditioning on interior data.

---

> > ### Author Response · Authors · 2025-11-21
> > **Rebuttal to Reviewer 2**
> >
> > > Missing references: Several relevant Gaussian process methods for PDEs should be cited (some suggestions have been provided by the reviewer). It will be better to position the work relative to existing Gaussian process techniques for PDEs (e.g., work by Andrew Stuart, Houman Owhadi, Mark Girolami, Philipp Hennig, Jon Cockayne). The novelty is unclear without reading deeper into the paper.
> >
> > Thank you for pointing this out. Our previous manuscript already contained a dedicated “state of the art” section and 90 references, including a range of Gaussian-process–based and probabilistic-numerics approaches to PDEs and inverse problems, but we agree that our positioning relative to key GP–PDE works could be made more explicit. We explicitly added several of the suggested approaches, including those mentioned here, so that the paper now has 99 references. We now clearly distinguish B-EPGP from collocation-based GP-PDE methods by emphasizing that our priors encode PDE + boundary constraints analytically rather than through pointwise conditioning. See the revised Section 4.
> >
> > > I really like line 90. The example helped me understand the problem being solved here. This is interesting: the functional form of the solution is known, but some parameters of this functional form are unknown. By imposing a Gaussian prior on these parameters, you effectively get a Gaussian process. This feels like a very smart construction of the kernel so that the solution space automatically aligns with the domain of the Gaussian process. I wouldn't be surprised if this performs better than PINNs because it exploits the particular PDE structure. Essentially, you've structured a specific Gaussian process that is engineered to overlap with the PDE solution space.
> > > Key Contribution (Lines 111-130)
> > > Line 111 is the real contribution and the key idea. Now it's very clear. Line 130 further explains the problem from line 111: the boundary conditions are encoded into the basis elements.
> >
> > We thank you for this comment and we now include this point in the abstract.
> >
> > > In Section 3.3, I didn't understand in what sense the method is "hybrid." Is it hybrid because you have both flat and curved pieces of boundary? If you're using data for the curved pieces, then you've somewhat lost your claim of things being exact. I think Sections 3.3 actually diminish the value proposition—you can plug in collocation points, but it doesn't integrate quite as well with the elegant solution you're proposing. This comment refers specifically to line 255.
> >
> > We agree that the hybrid variant mixes exact and collocation-based enforcement and will clarify this in the manuscript. Its purpose is to demonstrate that the B-EPGP basis for flat pieces can still be useful when the domain includes curved segments, by handling those segments via a small number of boundary points. We will explicitly state that (i) the core contribution is the fully analytic B-EPGP construction for flat / polygonal boundaries; (ii) the hybrid extension is optional and trades exactness for flexibility on curved geometries; and (iii) in our experiments the hybrid method still substantially improves boundary satisfaction and energy conservation over EPGP.

---

### Official Review · Reviewer_qz1s · 2025-10-31

**Soundness:** 3
**Presentation:** 3
**Contribution:** 2
**Rating:** 4
**Confidence:** 4

**Summary:**

This paper presents a new probabilistic framework, Boundary Ehrenpreis-Palamodov Gaussian Processes (B-EPGPs), for designing Gaussian Process (GP) priors that are constrained to satisfy a linear partial differential equations (PDEs) system with constant coefficients and a linear boundary condition system. The work builds upon existing "physics-constrained" methods like EPGP, which use the Ehrenpreis-Palamodov theorem to create priors whose realizations are exact solutions to the PDE system (represented as combinations of exponential-polynomial functions $e^{x \cdot z}$). However, existing methods have a hard time handling boundary conditions in a clean, analytical way. They often rely on conditioning the model on a large number of boundary data points—a strategy that’s both inefficient and quickly runs into the curse of dimensionality. The key idea behind B-EPGP is to build a new Gaussian Process basis that automatically satisfies the boundary conditions. For a given boundary, the method identifies groups of related frequencies—called “fibers,” $S_{z’}$—from the PDE’s characteristic variety. It then builds special linear combinations of basis functions, $\sum_{z \in S_{z’}} w_z e^{x \cdot z}$, that automatically satisfy the boundary conditions. The whole process is framed as a syzygy computation problem, which can be solved symbolically using Gröbner bases. It’s a neat and elegant way to handle what’s usually a messy part of the problem.

**Strengths:**

It provides a general, "physics-constrained" framework for a broad and critical class of PDE problems. By ensuring both the PDE and its boundary conditions are met exactly, B-EPGP offers a model that is more reliable, data-efficient, and computationally faster than methods that treat boundaries as soft penalties or data-fitting problems.

**Weaknesses:**

1. The algorithm for polygonal boundaries is based on an iterative process of the form "Repeat until S stabilizes." The authors say, "These steps in . . . do not generally terminate." While it is mentioned that for their examples it does terminate, the non-fulfillment of this important practical condition is not fully brought out. The conditions under which the process is guaranteed to terminate are not indicated.The infinite slab example is presented as a case of non-termination, but the solution is found via a "shortcut" (manually selecting a discrete Fourier basis $\xi \in \mathbb{Z}$) rather than as a direct output of the described iterative algorithm. This makes the polygonal algorithm feel less general and more "artisanal" than the very robust halfspace method.

2. The paper states that the Gröbner basis computations for finding fibers and syzygies have "negligible complexity in practice". This is justified by noting the computations are on the (small) operator equations, not the (large) data. While this holds for the single-equation PDEs (heat, wave) used in the examples, it is not clear if this optimism holds for large systems of coupled PDEs (e.g., linear elasticity or Maxwell's equations). The complexity of Gröbner basis algorithms can be severe, and the "negligible" claim should be better qualified by specifying the types of PDE systems for which it has been verified.

3. The paper extends the framework to inhomogeneous systems in a straightforward way: the solution is written as $u = u_p + v_{homo}$, where B-EPGP handles the homogeneous part, $v_{homo}$. However, this approach depends on being able to find a particular solution, $u_p$. While the authors describe a spectral method for doing this, their main examples use very simple cases where $u_p$ is just the forcing function $f$, which makes the demonstration a bit less convincing. This part of the contribution feels underdeveloped. If finding $u_p$ is itself a complex numerical task, it could become the new bottleneck and re-introduce numerical errors, which could undermine the "exact" nature of the B-EPGP framework.

**Questions:**

1. Could the authors be more precise about the termination conditions for the iterative algorithm for polygonal boundaries? Are there known classes of polygonal domains (e.g., convex polygons, or domains with angles that are rational multiples of $\pi$) for which termination is guaranteed? For the non-terminating case (like the infinite slab example), is the manual selection of a discrete basis the intended workflow, or can the algorithm be modified to output this basis directly?

2. Could the authors comment on the practical scalability of the symbolic pre-computation step (computing fibers and syzygies) when moving from single-equation PDEs to systems of coupled linear PDEs? For instance, have they explored the feasibility for systems like time-harmonic Maxwell's equations or linear elasticity?

3. The demonstration for inhomogeneous systems uses a known, simple particular solution $u_p$. How does the framework perform when $u_p$ is non-trivial and must be approximated using the proposed spectral method? Is there a risk that a numerical approximation of $u_p$ introduces significant errors, thereby negating the benefit of B-EPGP's exactness on the homogeneous part?

---

> ### Author Response · Authors · 2025-11-21
> **Rebuttal to Reviewer 1**
>
> We are very grateful to Reviewer 1 for their careful *attention to detail* and technically precise reading of our work, in particular regarding the termination of the polygonal algorithm, the practical complexity of Gröbner-basis computations, and the treatment of inhomogeneous systems. Your comments sharpened the distinction between finite and infinite reflection processes, pushed us to make the workflow in the slab example more transparent, and prompted us to better qualify our claims about symbolic pre-computation. We have revised the manuscript with these points in mind and thank you for helping us make the paper more precise and practically grounded.
>
> ## Regarding the termination of the boundary construction
>
> We thank the reviewer for raising the question of termination for the polygonal boundary algorithm and for pointing out that, in its present form, the slab example may appear artisanal. Our intention was precisely the opposite: the polygonal construction is a systematic extension of the halfspace method, and the slab example is meant to illustrate a situation where the reflection process is *infinite* by nature but still admits a concise closed-form description.
>
> In the revised paper we state clearly:
>
> - what “repeat until $S$ stabilizes” means in our construction (namely, closing a single bulk frequency $\hat z$ under the reflection group $G$ generated by the bounding halfspaces, i.e. $S = G \cdot \hat z$);
> - when this process *terminates* (finite reflection groups, e.g.\ rational wedges) versus when it yields an *infinite but discretely parametrizable* orbit (parallel boundaries in slabs/rectangles, leading to standard Fourier-type bases);
> - how the slab example arises *directly* from the same algorithm: we first construct the halfspace basis enforcing the boundary at $x = 0$, then enforce the boundary at $x = \pi$ via the algebraic condition $\sinh(b \pi) = 0$, whose solutions $b \in \sqrt{-1}\mathbb{Z}$ give exactly the usual sine modes $e^{\pm \sqrt{-1} j t}\sin(jx)$. The “shortcut” language only refers to solving this quantization condition in closed form, not to choosing an ad-hoc Fourier ansatz.
>
> From a practical perspective, the non-termination of the raw loop is not an issue in our experiments: for finite reflection groups we obtain a finite B–EPGP basis; for slabs and rectangles we obtain an explicitly parametrized infinite family of modes and then truncate, exactly as in standard spectral methods.
>
> All of these points are now discussed in detail (with worked examples for the wedge, slab, and rectangle) in the new Appendix C of the updated paper.
>
> ## Regarding Gröbner basis complexity for larger examples like linear elasticity or Maxwell's equations
>
> Thank you for pointing this out. We agree that Gröbner methods can be expensive in general. Our claim is meant in the restricted sense of the PDE systems considered in the paper, and we now back it up in Appendix B.2: also larger, coupled, and more complicated PDE systems like linear elasticity or Maxwell's equations terminate in fractions of a second (using Macaulay2, Singular or Maple) on standard laptops.
>
> ## Regarding the accuracy of our comment on inhomogeneous equations
>
> The solution spaces for both homogeneous and inhomogeneous equations are infinite dimensional. The strength of our method is that it handles this infinite dimensionality in a robust probabilistic framework. In particular, whenever we condition on data (e.g. nonzero boundary or initial conditions), errors are introduced. The merit of B-EPGP is that it does not introduce errors except the ones arising from unavoidable discretization of continuous initial/boundary conditions. Finding a particular solution $u_p$ is thus not a complex numerical task, as also evidenced by the explicit calculations in Appendix M.
> We are currently expanding the explication concerning the complexity of the calculation and will post a worked example with nontrivial $f$ and $g$ soon.
>
> [Edit: we have updated the document. A new example is in Appendix M.2. The errors introduced via the Fourier transform are in the order of $10^{-6}$.]

---

> > ### Author Response · Authors · 2025-11-28
> >
> > Dear Reviewer 1,
> > thank you again for your time and constructive feedback on our submission. We have carefully addressed your concerns in the rebuttal and added clarifications and experiments where possible. If you have a moment before the discussion phase ends, we would greatly appreciate it if you could take another look and let us know whether the changes resolve your main concerns.

---

### Author Response · Authors · 2025-11-21
**General comment**

We are grateful for all reviews, which clearly try to improve the paper. We have uploaded a slightly revised version, in which the improvements are marked by blue text color.

We give answers to the specific points by the reviewers below.

---

### Meta-Review · Area_Chair_QK2x · 2026-01-01

**Summary:**

This paper proposes Boundary Ehrenpreis--Palamodov Gaussian Processes (B-EPGPs), a novel probabilistic framework for constructing GP priors that satisfy both general systems of linear PDEs with constant coefficients and linear boundary conditions and can be conditioned on a finite data set. The Ehrenpreis--Palamodov theorem provides the functional form of the solution, but leaves parameters of this functional form unknown. An optimization finds these parameters and the authors provide an algorithm to modify this function form to satisfy boundary conditions. They explicitly construct GP priors for representative PDE systems with practical boundary conditions. Formal proofs of correctness are provided and empirical results demonstrating significant accuracy and computational resource improvements over state-of-the-art approaches. Reviewers agree that the work is theoretically strong, offering formal proofs and leveraging symbolic computation to encode boundary constraints analytically. However, some of the reviewers have also identified important weaknesses of this work. Specifically, they indicate that the algorithm for polygonal boundaries is based on an iterative process of the form "Repeat until S stabilizes". While it is mentioned that for their examples it does terminate, the non-fulfillment of this important practical condition is not fully brought out. This has been partially addressed in the rebuttal. The reviewers have also indicated that the method is restricted to linear PDEs with constant coefficients, which excludes nonlinear or variable-coefficient problems. This narrows applicability. Furthermore, the reviewers also point out that Section 3.3's hybrid approach (line 255) doesn't integrate well with the elegant exact solution framework. If data is needed for curved boundary pieces, the claim of exactness is compromised, which diminishes the value proposition. Some reviewers also found the exposition mathematically dense and lacking precision in definitions (e.g., basis vs. FAS, real vs. complex coefficients), making the paper hard to follow for non-specialists. This was addressed in the rebuttal. The reviewers also point out that the experiments focus on relatively simple geometries and PDEs. More complex or non-trivial domains would strengthen practical relevance (this was partially addressed in the rebuttal). Finally, the some reviewers also indicate that computational considerations are not discussed adequately (partially addressed in the rebuttal). Overall, while the method represents an advance in probabilistic PDE, there are still several areas of improvement of the paper. I therefore consider this contribution to be borderline.

**Reviewer Concerns:**

The rebuttal addressed some concerns such as clarifying the termination of the polygonal boundary algorithm, demonstrating efficiency for larger PDE systems, and adding examples for non-trivial problems. It also improved mathematical notation, explained optimization details, and expanded comparisons to related GP-PDE methods. However, some issues remain, including the method’s limited scope to linear PDEs with constant coefficients and the reliance on PDE-specific basis construction. Concerns about scalability in high dimensions (only partially commented in the rebuttal), the literature review (indicated to be included in the camera ready version), and limited experimental diversity also persist. Some of the reviewers indicate that the would increase their score after the rebuttal, but they could not do it.

**Reviewer Scores:**

Only reviewer r3kn may have increased their score during the rebuttal. In any case, I do not think that the increment would have been significant, which motivates my suggestion for a borderline paper.

---

### Decision · Program_Chairs · 2026-01-26

Reject